# ZSWIM4 regulates embryonic patterning and BMP signaling by promoting nuclear Smad1 degradation

Chengdong Wang [1,12], Ziran Liu [2,12], Yelin Zeng [1], Liangji Zhou [1], Qi Long [1], Imtiaz Ul Hassan [1], Yuanliang Zhang [3], Xufeng Qi [4], Dongqing Cai [4], Bingyu Mao [5,6], Gang Lu [1], Jianmin Sun [7], Yonggang Yao [5,6], Yi Deng [8], Qian Zhao [3], Bo Feng [1], Qin Zhou [9], Wai Yee Chan [1,10,11] & Hui Zhao [1,10,11 ✉]

## Abstract

**The dorsoventral gradient of BMP signaling plays an essential role in embryonic patterning.** *Zinc Finger SWIM-Type Containing 4* (*zswim4*) **is expressed in the Spemann-Mangold organizer at the onset of** *Xenopus* **gastrulation and is then enriched in the developing neuroectoderm at the mid-gastrula stages. Knockdown or knockout of** *zswim4* **causes ventralization. Overexpression of** *zswim4* **decreases, whereas knockdown of** *zswim4* **increases the expression levels of ventrolateral mesoderm marker genes. Mechanistically, ZSWIM4 attenuates the BMP signal by reducing the protein stability of SMAD1 in the nucleus. Stable isotope labeling by amino acids in cell culture (SILAC) identifies Elongin B (ELOB) and Elongin C (ELOC) as the interaction partners of ZSWIM4. Accordingly, ZSWIM4 forms a complex with the Cul2-RING ubiquitin ligase and ELOB and ELOC, promoting the ubiquitination and degradation of SMAD1 in the nucleus. Our study identifies a novel mechanism that restricts BMP signaling in the nucleus.**

**Keywords** Zswim4; BMP; Spemann–Mangold Organizer; *Xenopus*; Cullin RING Ubiquitin Ligase
**Subject Categories** Development; Post-translational Modifications & Proteolysis; Signal Transduction

## Introduction

Embryonic patterning is a stepwise process requiring precise temporal and spatial regulation of multiple signaling pathways, such as Wnt, Nodal, and BMP signaling (Baker et al, 1999; De Robertis and Moriyama, 2016; Niehrs, 2012; Zinski et al, 2018). The coordination of these pathways leads to the activation of cohorts of genes at the appropriate time and position in embryos and governs cell fate specification. The formation and function of the Spemann–Mangold organizer (SMO) during gastrulation and subsequent neural induction are two distinct but closely related landmarks during embryonic development. These processes received particular attention because of their unique roles in the embryonic induction (Harland and Gerhart, 1997; Niehrs, 1999).

The BMP signaling pathway plays an essential role in dorsoventral patterning during gastrulation (Bier and De Robertis, 2015). The activation of BMP signaling is triggered by the binding of BMP ligands to their receptors, which induces the phosphorylation of receptor-regulated Smads (R-Smad 1/5/8 or 1/5/9); the phosphorylated R-Smads interact with Smad4 to form a complex, which then enters the nucleus and initiate transcription of the target genes (Guglielmi et al, 2021; Tsukamoto et al, 2014). A dorsoventral gradient of BMP signaling is established by the cooperation of BMP ligands produced on the ventral side and BMP antagonists, such as Chordin and Noggin, secreted from the SMO (Hemmati-Brivanlou and Thomsen, 1995; Sasai et al, 1994; Smith and Harland, 1992). A high level of BMP signal on the ventral side induces the epidermis and ventral mesoderm formation via activating *xhox3*, *vent1*, while intermediate levels of BMP signal in the lateral marginal zone and a very low level of BMP in the dorsal side favors the formation of neuroectoderm and dorsal mesoderm (Dosch et al, 1997; Gawantka et al, 1995;

[1]Key Laboratory for Regenerative Medicine, Ministry of Education, School of Biomedical Sciences, Faculty of Medicine, The Chinese University of Hong Kong, Hong Kong SAR, China. [2]Qingdao Municipal Center for Disease Control and Prevention, 266033 Qingdao, Shandong, China. [3]State Key Laboratory of Chemical Biology and Drug Discovery, Department of Applied Biology and Chemical Technology, The Hong Kong Polytechnic University, Hong Kong SAR, China. [4]Key Laboratory of Regenerative Medicine of Ministry of Education, Department of Developmental & Regenerative Biology, Jinan University, 510632 Guangzhou, Guangdong, China. [5]Key Laboratory of Animal Models and Human Disease Mechanisms, Kunming Institute of Zoology, Chinese Academy of Sciences, 650223 Kunming, Yunnan, China. [6]Kunming Institute of Zoology - The Chinese University of Hong Kong (KIZ-CUHK) Joint Laboratory of Bioresources and Molecular Research of Common Diseases, Chinese Academy of Sciences, Kunming, China. [7]Department of Pathogen Biology and Immunology, School of Basic Medical Sciences, Ningxia Medical University, No. 1160 Shengli Street, 750004 Yinchuan, China. [8]Department of Biology, Guangdong Provincial Key Laboratory of Cell Microenvironment and Disease Research, and Shenzhen Key Laboratory of Cell Microenvironment, Southern University of Science and Technology, 518055 Shenzhen, China. [9]School of Basic Medical Sciences, Harbin Medical University, 150081 Harbin, China. [10]Kunming Institute of Zoology - The Chinese University of Hong Kong (KIZ-CUHK) Joint Laboratory of Bioresources and Molecular Research of Common Diseases, The Chinese University of Hong Kong, Hong Kong SAR, China. [11]Hong Kong Branch of CAS Center for Excellence in Animal Evolution and Genetics, The Chinese University of Hong Kong, Hong Kong SAR, China. [12]These authors contributed equally: Chengdong Wang, Ziran Liu. ✉E-mail: zhaohui@cuhk.edu.hk

Hemmati-Brivanlou and Thomsen, 1995). In addition to the level of available BMP ligands, BMP signaling is also regulated by controlling the stability of signaling components. Smurf1 and Smurf2, two HECT type E3 ubiquitin ligases, were reported to induce the ubiquitination and degradation of R-Smads and BMP type I receptors during *Xenopus* embryonic development (Ebisawa et al, 2001; Koganti et al, 2018; Murakami et al, 2003; Zhang et al, 2001; Zhu et al, 1999). ZC4H2 was reported to stabilize Smad1 by antagonizing its ubiquitination by Smurf (Ma et al, 2017).

Kctd15 plays important roles in *Xenopus* and zebrafish embryonic development (Dutta and Dawid, 2010; Heffer et al, 2017). We identified *Zinc finger SWIM-type containing 4* (*zswim4*) that is down-regulated by loss of Kctd15 (Wong et al, 2016). The cellular function of Zswim4 and its roles during embryonic development are largely unknown. We found that Zswim4 is a maternal factor, and then specifically expressed in the SMO at the onset of gastrulation and during the formation of the neural plate. Gain-of-function and loss-of-function assays consistently suggest that Zswim4 restricts the ventrolateral mesoderm formation and maintains the neural tissue. Mechanistically, Zswim4 physically interacts with Smad1 and the Cul2-RING ubiquitin ligase complex to induce the ubiquitination and degradation of Smad1, leading to the attenuation of BMP signaling outputs. Our study reveals a new regulatory mechanism of BMP signaling by Zswim4.

## Results

### Zswim4 is expressed in the SMO and shows nuclear localization

We previously identified *zswim4* that responds to the activation of Kctd15 (Wong et al, 2016). The Zswim4 harbors a nuclear localization signal at the N terminus and a SWIM domain named after SWI2/SNF and MuDR transposases. The SWIM domain is followed by a cut8/STS1 domain which is related to the proteasomes in the nucleus (Fig. 1A; Grzenda et al, 2009; Makarova et al, 2002; Witteveen et al, 2016). The zinc finger SWIM domain-containing protein family is a novel protein family. Although Zswim8 has been reported to regulate the degradation of targeted microRNA, and functions as a ubiquitin ligase (Han et al, 2020; Shi et al, 2020), the functions of this protein family remain largely unknown. Protein alignment indicates that Zswim4 is highly conservative among different species (Figure EV1A). The *X. laevis* Zswim4 shares 69.9% and 74.3% identities at the protein level with the human and mouse homologs, respectively (Figure EV1B).

We first examined the temporal and spatial expression patterns of *zswim4*. The temporal expression pattern revealed by quantitative RT-PCR showed that *zswim4* has maternal expression which is high and maintained at a constant level before gastrulation (Fig. 1B). However, the expression of *zswim4* is sharply decreased after gastrulation (Fig. 1B). Whole mount in situ hybridization (WISH) showed a high steady-state level of *zswim4* expression at the cleavage and blastula stages, with enriched signals in the animal hemisphere (Figure EV1C,D). At the onset of gastrulation, *zswim4* expression in the animal pole was sharply reduced, and *zswim4* signals exhibited a gradient pattern in the marginal zone with intensive signals in the dorsal blastopore lip and the SMO (Figs. 1C,D and EV1E,F), and were then enriched at the dorsal

ectoderm at mid-gastrula stages (Figure EV1G). Its expression was detected in the anterior edge of the neural plate at neural stages (Figure EV1H), suggesting a possible role in neural differentiation and brain formation. *Zswim4* expression was detected in the lens, brain, and some cranial nerves in the tailbud stage (Figure EV1I,J). Sections of the stained embryos also revealed *zswim4* expression in in the foregut, neural tube, and the region surrounding the notochord (Figure EV1K,L). Cryosection and immunofluorescence staining were performed on *Xenopus* embryos of stage 11 with Zswim4 antibody. Zswim4 protein is enriched in the dorsal blastopore lip and the SMO with nuclear localization (Fig. 1D,E). It is notable that we also detected Zswim4 signals in the endoderm and ectoderm, although the signal density is less compared to that in SMO. As a control, we conducted immunofluorescence staining using rabbit IgG instead of the Zswim4 antibody. Importantly, no detectable signals were observed when using rabbit IgG, indicating the specificity of Zswim4 antibody (Figure EV1M). Cellular localization of ZSWIM4 was examined in Hela cells transfected with *ZSWIM4-FLAG*. Immunostaining showed that ZSWIM4 is also predominantly localized in the nucleus (Fig. 1F).

We next performed animal cap assays to study the regulation of *zswim4* transcription. We injected either *bmp4*, *fgf8*, or *wnt3a* mRNA into both blastomeres of embryos at the two-cell stage. The animal caps were dissected at stage 9 and cultured for 2 h. The expression of *zswim4* was examined by RT-PCR. Indeed, the overexpression of either *bmp4*, *fgf8*, or *wnt3a* suppressed the expression of *zswim4* (Fig. 1G). The injections were effective, as the expression of *epidermal keratin* (*epiker*) and *xbra* for *bmp4*, the expression of *xbra* for *fgf8*, and *nr3* for *wnt3a* were successfully induced (Fig. 1G; Agius et al, 2000). We also injected *chordin* mRNA into embryos at the two-cell stage. Similar to *sox2*, which is a pan neural marker gene (Mizuseki et al, 1998), the *zswim4* expression is increased upon overexpression of *chordin* in animal cap assay (Fig. 1H). The response of *zswim4* transcription is consistent with its expression in the dorsal blastopore lip and neural plate during embryonic development.

### Dysregulation of *zswim4* disturbs anterioposterior axis formation

To investigate Zswim4 function during early embryonic development, *zswim4* mRNA was injected into *X. laevis* embryos at the two-cell stage. Injection of *zswim4* mRNA at 100 pg per embryo resulted in substantial embryonic mortality by the neurula stage. Thus, we injected *zswim4* mRNA with three low doses (12.5 pg/embryos, 25 pg/embryo and 50 pg/embryo). When developing to the tailbud stages, the injected embryos showed an apparent disturbance of posterior body axis formation with evident shortening or absence of the tail, and this phenotype, which was categorized as Dorsoanterior index (DAI) 6 (Kao and Elinson, 1988), exhibited a dose-dependent manner (Fig. 2A,B). However, the head region remained largely unaffected at these dosages (Fig. 2A).

We designed two antisense morpholino oligonucleotides (MO, zswim4MO1 and zswim4MO2) for *zswim4* knockdown experiments in *X. tropicalis* embryos. Zswim4MO1 is a translation blocking MO targeting the 5' untranslated region of *zswim4*. Western blot showed that zswim4MO1 effectively blocked the translation of *zswim4-FLAG* harboring the zswim4MO1 binding

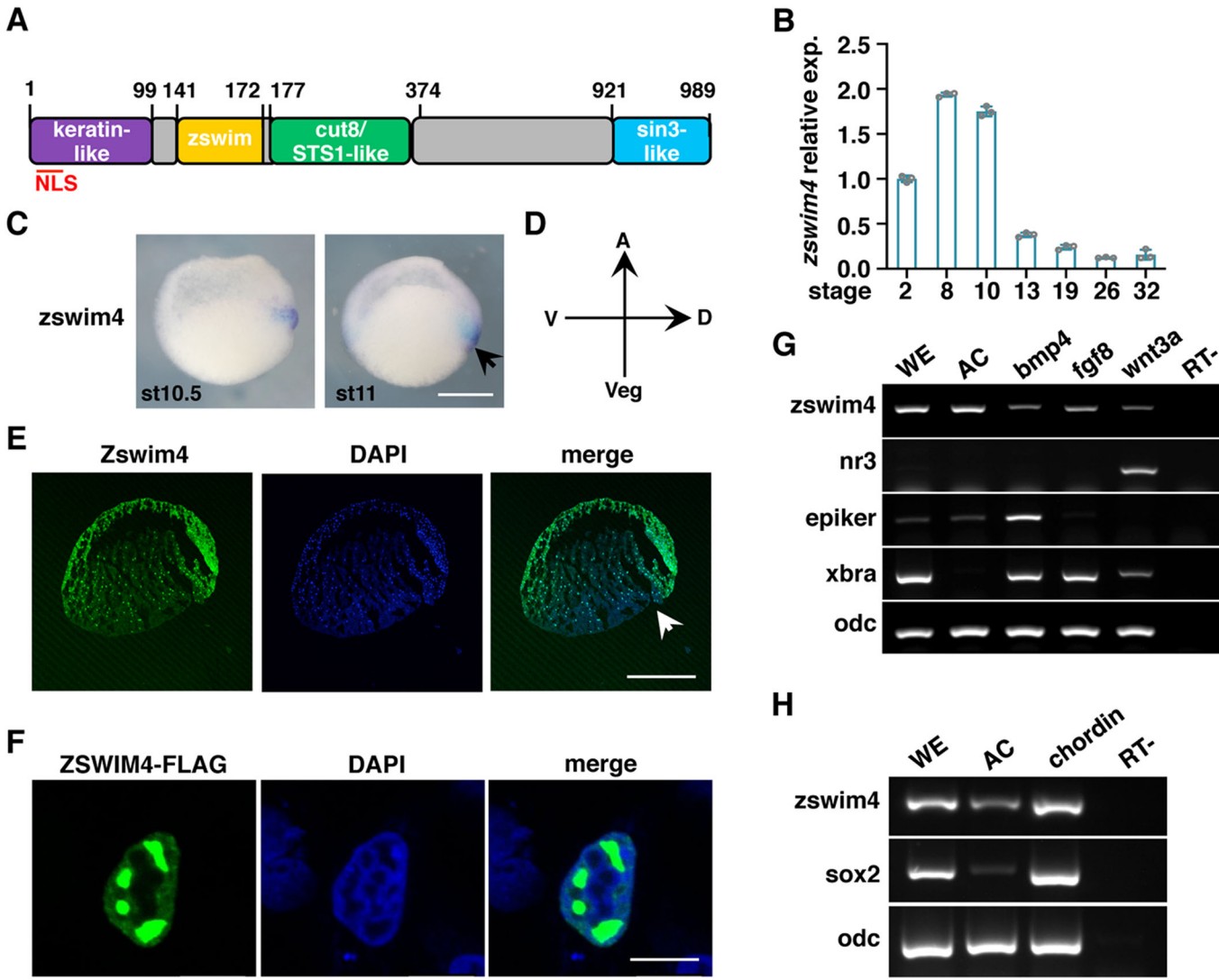

**Figure 1. The expression of zswim4 in developing embryos and its cellular localization.**

(A) A schematic diagram illustrating the different protein domains in the Zswim4. (B) Temporal expression of *zswim4* in *X. laevis* embryos at the indicated stages. *Ornithine decarboxylase* (*odc*) was used as an internal standard. *n* = 3. (C) Spatial expression pattern of *zswim4* in bisected *X. laevis* embryos revealed by whole mount in situ hybridization. The black arrow indicates the signals on the dorsal side at stage 11. Scale bar = 500 μm. (D) Arrows indicating the orientation of the embryos in (C, E). (E) Immunofluorescence staining showing the localization of Zswim4 protein in *X. laevis* embryos of stage 11. The white arrow indicates the dorsal blastopore. Scale bar = 500 μm. (F) Confocal images of Hela cells transfected with *ZSWIM4-FLAG*. Scale bar = 5 μm. (G, H) Semi-quantitative RT-PCR analysis of expression of *zswim4* and the indicated genes in *X. laevis* animal caps dissected from embryos injected with mRNAs of either *bmp4* (300 pg/embryo), *fgf8* (100 pg/embryo), *wnt3a* (300 pg/embryo), or *chordin* (500 pg/embryo), respectively. WE, uninjected whole embryo; AC, uninjected animal caps; RT-, without reverse transcriptase. *n* = 2. One representative image is shown. Data information: *n* indicates biological replicates. Error bars show mean ± standard deviation (SD). Source data are available online for this figure.

---

sequence (Figure EV2A) in *X. tropicalis*. The zswim4MO2 is an mRNA splicing MO designed to target the junction region of *zswim4* intron 1 and exon 2, which causes mis-splicing of *zswim4* pre-mRNA by joining exon 1 to exon 3 directly (Figure EV2B). A pair of primers bridging exon 1 and exon 3 was designed to test the efficiency of zswim4MO2 (Figure EV2B). RT-PCR results showed that the mis-spliced band of *zswim4* (200-bp band) appeared in the sample from the zswim4MO2 injected embryos (10 ng/embryo), while the level of wild-type band of *zswim4* (400-bp band) was evidently reduced compared with that in the uninjected embryo (Figure EV2C), suggesting the high efficiency of zswim4MO2. Both

bands were confirmed by sequencing (Figure EV2D,E). We also examined the endogenous Zswim4 protein in *zswim4* morphant. Both zswim4MO1 and zswim4MO2 were found to reduce the expression of endogenous Zswim4 protein (Figure EV2F,G). This result provided additional evidence supporting the specificity of the Zswim4 antibody in recognizing *Xenopus* Zswim4. The injection of zswim4MO1 or zswim4MO2 caused a consistent phenotype of defective head formation, shortened body axis, and enlarged ventral region, reflecting a DAI of 4 (Fig. 2C). The defective head formation was confirmed by the reduced expression of the brain marker, *otx2* (Fig. 2D). These phenotypes showed a

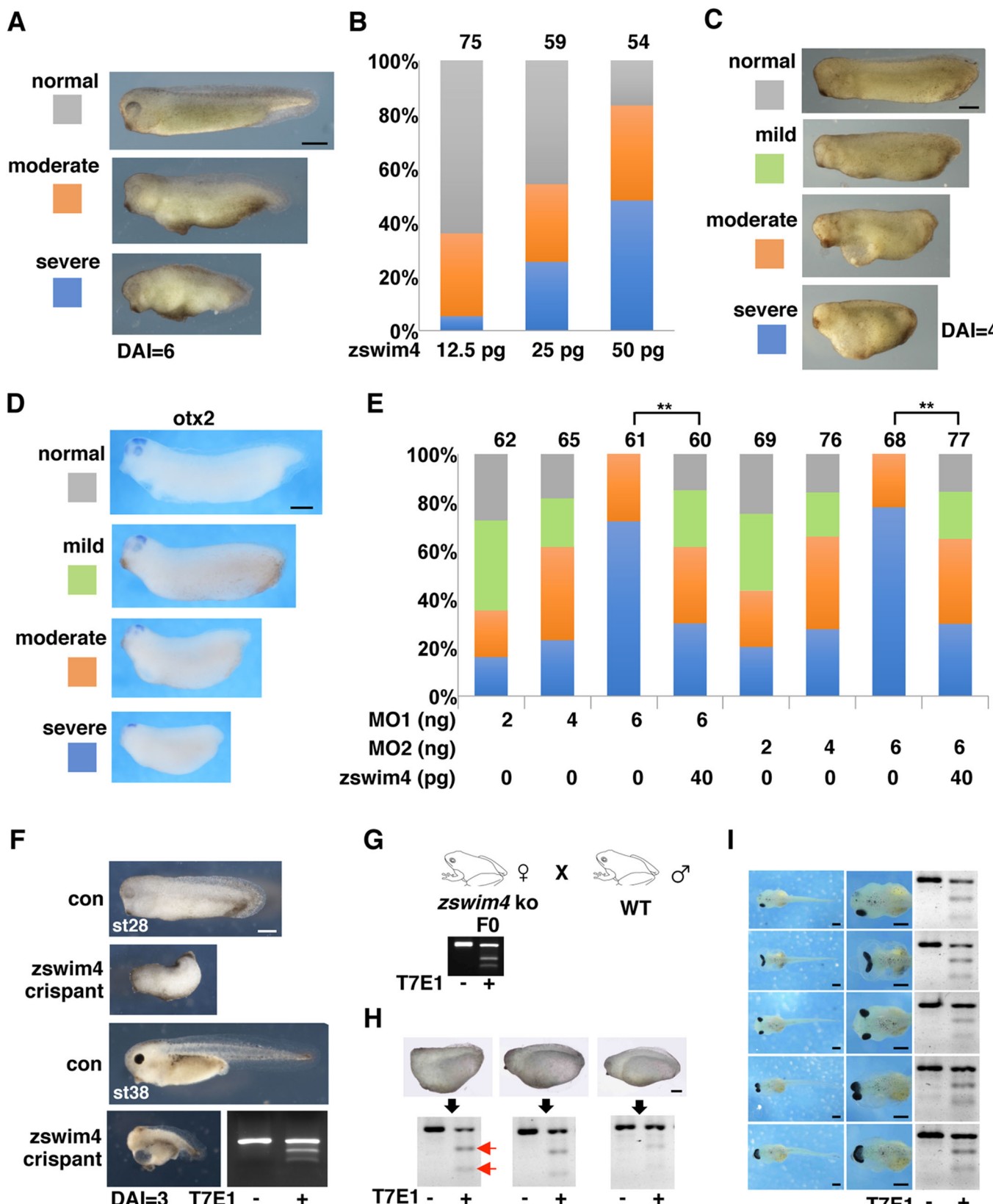

**Figure 2. Dysregulation of *zswim4* disturbs axis formation during embryonic development.**

(A, B) Categories of defects in the *X. laevis* embryos injected with increasing doses of *zswim4* mRNA at the two-cell stage. Scale bar = 500 µm. The numbers on the top indicate the total number of embryos. n = 3. (C–E) Categories of defects in the *X. tropicalis* embryos injected with different doses of either zswim4MO1, zswim4MO2, or the combination of *zswim4* mRNA with MOs. The representative images of each phenotype category were shown in (C). The expression of *otx2* was examined in these defective embryos (D). Scale bar = 250 µm. The numbers on the top in (E) indicate the total number of embryos. n = 3. (F) Phenotype of *X. tropicalis* embryos injected with Cas9 protein (0.5 ng/embryo) and four sgRNAs (total 200 pg/embryo) targeting *zswim4* (35 of 57 injected embryos at stage 28; 23 of 49 injected embryos at stage 38. n = 2, biological replicates). Scale bar = 250 µm. T7E1 assays were performed to confirm the gene disruption. (G–I) The F0 generation of the *X. tropicalis zswim4* mutant frogs with high gene disruption efficiency were crossed with wild-type male frogs (G). Anterior and body axis defects were observed in the offspring embryos at stage 28 (H) or stage 43 (I). Scale bar = 250 µm. Genomic DNA was extracted from individual embryos for the T7E1 assay. The DNA gel electrophoresis results showed the T7E1 assay using DNA extracted from the corresponding larva on the top (H) or on the left (I). Data information: n indicates biological replicates. Error bars show mean ± standard deviation (SD). Statistical analysis was performed using the chi-squared test for (E). **$p < 0.01$. Source data are available online for this figure.

dose-dependent manner (Fig. 2E). Importantly, these adverse effects could be partially rescued by co-injection with *zswim4* mRNA without the 5' untranslated region (Fig. 2E), suggesting that the phenotype caused by zswim4MOs is specific.

CRISPR/Cas9 mediated genomic editing was used to introduce mutations to the *Xenopus zswim4* gene. We first designed four sgRNAs targeting *zswim4* (sgRNA1–4 in Figure EV3A) and co-injected them with Cas9 protein into *X. tropicalis* embryos (Wu et al, 2018). T7E1 assay was used to assess the gene disruption efficiency (Fig. 2F). In more than half of the embryos of the tail-bud stage, CRISPR/Cas9 mediated disruption of *zswim4* caused short tail and trunk region, as well as loss of head structures, including eyes and brain, which was categorized as DAI 3 (Fig. 2F). This phenotype was highly similar to the phenotype induced by the MOs (Fig. 2C). Taken together, these results indicate the critical roles of Zswim4 in body axis formation and head development.

We further chose two sgRNAs targeting the first exon of *X. tropicalis zswim4* (sgRNA4 and 5 in Figure EV3A) to generate a *zswim4* mutant line. The two sgRNAs were injected separately with Cas9 protein into the *Xenopus* embryos at the one-cell stage (100 pg sgRNA and 0.5 ng Cas9 protein/embryo). After confirming the gene disruption by T7E1 assay and Sanger sequencing, the injected embryos and their wild-type siblings were raised into adults (Figure EV3B–D). Female F0 frogs (*zswim4F0*) were then crossed with wild-type male frogs (Fig. 2G). In two independent crosses, approximately 26% of the offspring (33 out of 126) showed a ventralization phenotype, i.e., a shortened body axis, an enlarged ventral side, and suppression of the head (Fig. 2H). This phenotype was also observed in zswim4MO- or CRISPR/Cas9-injected embryos (Fig. 2C,F). Most of the embryos showing the phenotype died at the later stages, and the larvae that survived had relatively normal body axis, but with cyclopia, or various degrees of eye field split failure (Fig. 2I). Three representative embryos at stage 28 and five larvae at stage 42 were collected for the T7E1 assays. All of these embryos harbor mutated alleles and should be heterozygous (Fig. 2H,I).

After raising the mutant embryos to adults, we obtained the F1 generation of *zswim4* mutant *X. tropicalis* frogs containing a 10-bp deletion, *zswim4*$^{+/-}$ (Figure EV3E). The mutant allele of *zswim4*$^{+/-}$ contains a premature stop codon and encodes a truncated form of Zswim4 without most of its protein domains, including the cut8 and zswim domains (Figure EV3E). We further crossed two *zswim4*$^{+/-}$ mutants, and the offspring was named *zswim4F2*, including wild-type, *zswim4*$^{+/-}$ and *zswim4*$^{-/-}$ embryos. Maternal zygotic *zswim4*$^{-/-}$ mutant embryos (*MZzswim4*$^{-/-}$) were also obtained by crossing two *zswim4*$^{-/-}$ frogs. However, we observed no obvious morphological defects in the *zswim4F2* embryos or *MZzswim4*$^{-/-}$

embryos until the tailbud stages. We reason that other members of the Zswim protein family compensate for the loss of the Zswim4, as CRISPR/Cas9 may induce genetic compensation during the process of the establishment of the zebrafish mutant line (Ma et al, 2019). To confirm this, we examined the expression of homologous genes in the Zswim family with quantitative RT-PCR. Indeed, the expression of *zswim5*, *zswim6*, and *zswim7* was increased approximately two-fold in *MZzswim4*$^{-/-}$ embryos compared with wild-type embryos (Fig EV4A). In *zswim4F2* embryos, all of the zswim genes except *zswim9* exhibited increased expression levels compared with their levels in wild-type embryos, with *zswim7* showing an eight-fold increase (Figure EV4A). Due to the strong increase in *zswim7* expression, we then performed WISH to examine its expression in *zswim4F2* embryos. Compared with the weak expression in wild type embryos (Hassan et al, 2023), *zswim4F2* embryos showed strong signals in the dorsal region, while the elevated *zswim7* expression was not observed in *zswim4* crispants (Figure EV4B,C). These data suggest that the genetic compensation accounted for the phenotype discrepancy between F0 embryos and F2 embryos of the *zswim4*-knockout line. To assess the contribution of *zswim7* in the genetic compensation of *zswim4F2*, MO targeting *zswim7* (z7MO) was injected into *zswim4F2* embryos. Knockdown of *zswim7* in *zswim4F2* evidently increased the ratio of defective embryos which showed shortened and curved tail (Figure EV4D,E). In line with that, the defects in *zswim4* crispants were partially rescued by co-injection of *zswim7* mRNA in a dose-dependent manner (Fig EV4F,G). These results showed that the up-regulated *zswim7* did contribute to the genetic compensation in *zswim4F2*. Such genetic compensation was not observed in F0 embryos simultaneously injected with Cas9 protein and four sgRNAs targeting *zswim4* (Figs. 2F and EV4A).

## Zswim4 restricts the ventrolateral mesoderm and promotes neural formation by inhibiting BMP signaling

As Zswim4 affects body axis formation and is expressed in the SMO and the forming neural plate, we sought to determine whether the marker genes for the ventral mesoderm and neural ectoderm were affected by the dysregulation of Zswim4. Whole-mount in situ hybridization results showed that overexpression of *zswim4* severely reduced the expression regions of *sizzled*, *vent1*, *vent2*, and *myod* in *X. laevis* embryos injected with *zswim4* and *lacZ* mRNAs into one ventral blastomere at the four-cell stage (Fig. 3A–D). The expression of neural ectoderm marker gene, *sox2*, was decreased after the knockdown of *zswim4* with zswim4MO1 (Fig. 3E). The anterior endomesoderm marker, *cerberus*, and the prechordal plate marker, *gsc*, were repressed by

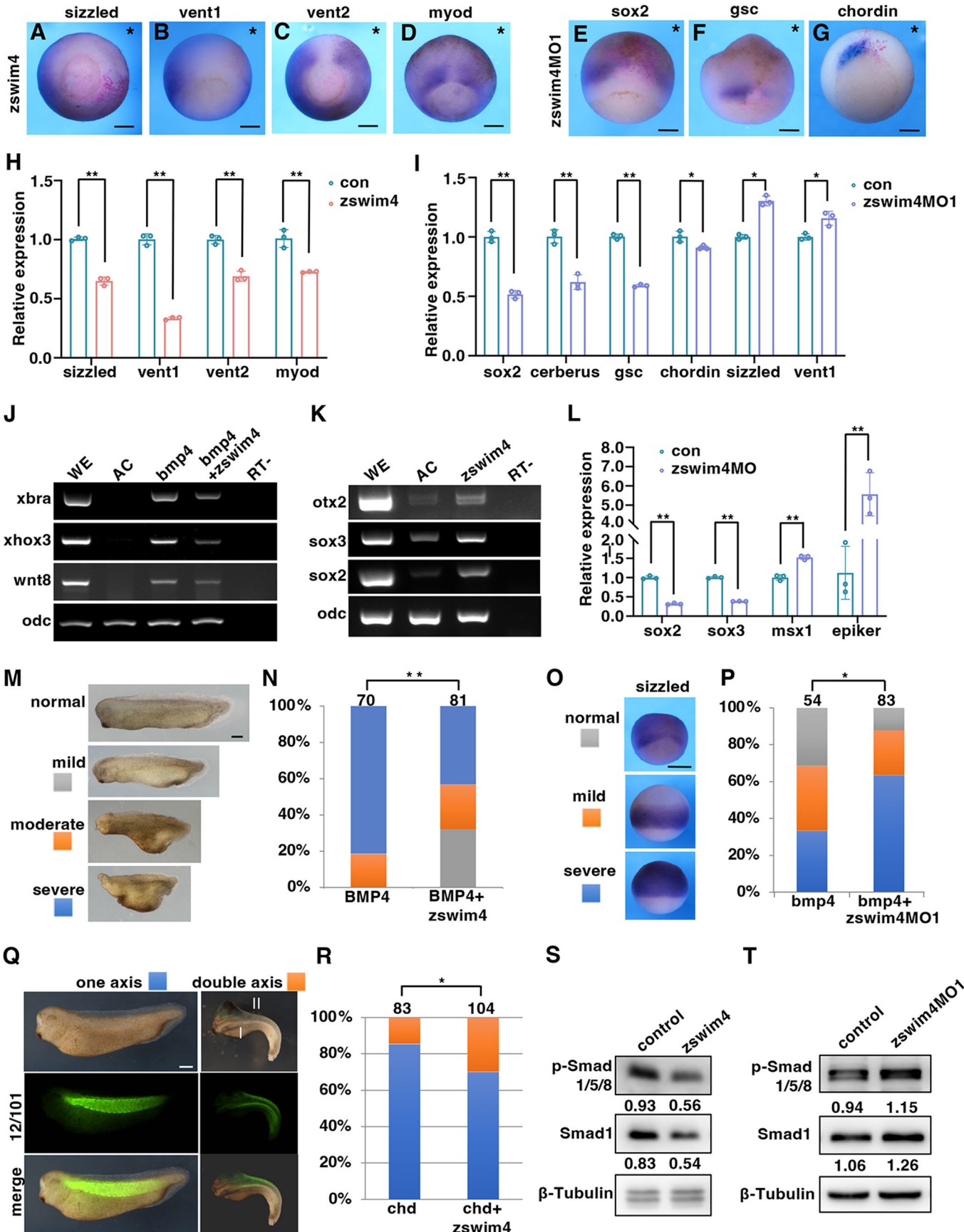

**Figure 3. Zswim4 restricts ventral mesoderm formation and promotes neural induction by inhibiting BMP signaling.**

(A–D) Overexpression of *zswim4* reduced the signals of *sizzled* (47 of 60 embryos), *vent1* (45 of 62 embryos), *vent2* (50 of 56 embryos), and *myod* (46 of 57 embryos), as revealed by in situ hybridization. *n* = 2. Scale bar = 250 μm. The *lacZ* mRNA (100 pg/embryo) and *zswim4* mRNA (200 pg/embryo) were co-injected into one ventral blastomere of *X. laevis* embryos at the four-cell stage. Red X-gal staining was employed to trace the injected side which is indicated by asterisks. (E–G) Zswim4MO1 (2 ng/embryo) and *lacZ* mRNA (100 pg/embryo) were co-injected into the one dorsal blastomere of *X. tropicalis* embryos at the four-cell stage. Knockdown of *zswim4* decreased the levels of *sox2* (34 of 56 embryos), *gsc* (42 of 61 injected embryos), and *chordin* (20 of 54 embryos). *n* = 2. Scale bar = 100 μm. Red X-gal staining was performed to trace the injected side which is indicated by asterisks. (H, I) Quantitative RT-PCR was performed to examine the expression of indicated markers in *X. laevis* embryos injected with *zswim4* mRNA (200 pg/embryo) or *X. tropicalis* embryos injected with zswim4MO1 (2 ng/embryo). *n* = 3. (J–L) Animal cap assays of *X. laevis* embryos were performed and semi-quantitative (J, K) or quantitative (L) RT-PCR were performed to examine indicated markers. For semi-quantitative RT-PCR (J, K), *n* = 2, and one representative image is shown. For qPCR (L), *n* = 3. (M, N) Categories of ventralization of the *X. laevis* embryos injected with *bmp4* mRNA (300 pg/embryo) and *zswim4* mRNA (150 pg/embryo) (M), and the quantification of the phenotype (N). Scale bar = 250 μm. The numbers on the top of the columns (N) indicate the total number of embryos. *n* = 3. (O, P) WISH showing different levels of *sizzled* expression in *X. tropicalis* embryos injected with *bmp4* mRNA (60 pg/embryo) and zswim4MO1 (4 ng/embryo). Scale bar = 200 μm. The numbers on the top of the columns (P) indicate the total number of embryos. *n* = 3. (Q, R). Either *chordin* mRNA (30 pg/embryo) alone or the mixture of *chordin* and *zswim4* mRNAs (200 pg/embryo) were injected into the ventral side of the *X. laevis* embryos at the four-cell stage. When developing to the tailbud stage, the embryos were collected for immunofluorescence staining with the 12/101 antibody that can label somites. The embryos with or without the secondary body axis were scored. Scale bar = 250 μm. The numbers on the top of the columns (R) indicate the total number of embryos. *n* = 3. (S, T) *X. laevis* embryos injected with *zswim4* mRNA (200 pg/embryo) or *X. tropicalis* embryos injected with zswim4MO1 (4 ng/embryo) were collected at stage 11 to examine p-Smad1/5/8 and total Smad1. β-Tubulin was used as the loading control. *n* = 2. Data information: *n* indicates biological replicates. Error bars show mean ± standard deviation (SD). Statistical analysis was performed using an unpaired Student's *t* test for (H, I, L), and chi-squared test for (N, P, R). *$p < 0.05$, **$p < 0.01$. Source data are available online for this figure.

the knockdown of *zswim4* (Appendix Fig. S1 and Fig. 3F), which may explain the defective head formation in the *zswim4* morphant (Bouwmeester et al, 1996; Sander et al, 2007). The reduced expression of *chordin*, the SMO marker, was only observed in a small portion of embryos with *zswim4* knockdown (Fig. 3G). The changes in the expression of these markers were confirmed by quantitative RT-PCR (Fig. 3H,I). Meanwhile, the ventral mesoderm marker genes, *sizzled* and *vent1*, were increased (Fig. 3I). Collectively, these results indicate that Zswim4 is required to restrict the ventral mesoderm formation and promote neural fate in the early gastrula.

As the BMP gradient along the dorsoventral axis plays an important role in ventral mesoderm patterning and neural induction, we next examined whether Zswim4 regulates embryonic development by antagonizing BMP signaling. To test whether Zswim4 inhibits BMP signaling transduction, we injected *bmp4* mRNA (300 pg/embryo) into both blastomeres of embryos at the two-cell stage and then performed animal cap assays. The activation of BMP signaling was confirmed by the up-regulation of *xhox3*, *xbra*, and *wnt8*, three BMP-responsive genes (Fig. 3J; Kirmizitas et al, 2014). The expression levels of these genes were attenuated by the co-injection of a high dose of *zswim4* mRNA (200 pg/embryos) (Fig. 3J), suggesting the suppression of BMP signaling by *zswim4*. Consistent with this observation, the overexpression of *zswim4* (200 pg/embryos) alone in the animal cap induced the expression of neural marker genes, *sox2*, *sox3*, and *otx2* (Fig. 3K), suggesting that the endogenous BMP signal was also attenuated by *zswim4*. Animal cap assays were also performed using embryos injected with MO targeting *X. laevis zswim4* (zswim4MO, 20 ng/embryo). Quantitative RT-PCR showed knockdown of *zswim4* repressed neural marker genes, *sox2* and *sox3*, and simultaneously enhanced the expression of BMP marker genes, *msx1* and *epiker* (Fig. 3L). These findings indicate that Zswim4 can repress BMP signaling in the animal caps. When examining the whole embryos, *bmp4*-induced ventralization was attenuated by co-injection with *zswim4* mRNA, suggesting again an inhibition role of Zswim4 on BMP signaling (Fig. 3M,N). To examine whether endogenous Zswim4 also plays a role in antagonizing BMP signal, we overexpressed *bmp4* in embryos, resulting in 33% of the injected

embryos with strong expression of *sizzled*. After co-injection of zswim4MO1 with *bmp4* mRNA, 57% of the injected embryos showed strong *sizzled* expression (Fig. 3O,P). These results suggested that endogenous Zswim4 restricts ventral mesoderm formation by antagonizing BMP signaling. Chordin antagonizes the BMP signaling pathway, and injection of *chordin* induces a secondary axis when ventral blastomeres are targeted (Plouhinec et al, 2013; Sasai et al, 1994). Immunofluorescence staining was performed on embryos of the tail-bud stage with antibody against 12/101, a skeletal muscle marker, to reveal the body axis (Fig. 3Q). 14% (12 of 83) of the embryos injected with *chordin* mRNA had a secondary axis, while after co-injection of *zswim4* mRNA and *chordin* mRNA, 30% (31 of 104) of the embryos had a secondary axis (Fig. 3R). Phosphorylated Smad1/5/8 (p-Smad1/5/8) is an essential mediator of BMP signal activation. Overexpression of *zswim4* in the whole embryo (200 pg/embryos) apparently reduced the level of p-Smad1/5/8 and total Smad1 (Figs. 3S and EV5A). Conversely, the knockdown of *zswim4* caused a moderate increase in total Smad1 levels and an elevation of p-Smad1/5/8 levels (Figs. 3T and EV5B). These results indicate that Zswim4 inhibits BMP signaling activity in both whole embryos and animal caps.

## ZSWIM4 attenuates BMP signaling in the HEK293T cell line

To examine whether the inhibitory role of Zswim4 on BMP signaling is conserved in mammalian cells, we performed luciferase assays in HEK293T cells transfected with a BMP responsive element (BRE)-luciferase reporter. BMP4-induced luciferase activity was inhibited by transfected *ZSWIM4* in a dose-dependent manner (Fig. 4A). Similarly, the expression of *ID1* and *ID2* induced by BMP treatment was also down-regulated in *ZSWIM4* transfected cells (Fig. 4B,C). Thus, overexpression of *ZSWIM4* inhibits BMP signaling in HEK293T cells. We next knocked down endogenous *ZSWIM4* through the transfection of siRNAs targeting human *ZSWIM4*. All three siRNAs can effectively attenuate *ZSWIM4* expression, resulting in approximately a 60% reduction in *ZSWIM4* mRNA levels compared with the levels in control cells (Appendix Fig. S2). Knockdown of *ZSWIM4* promoted BMP4-induced

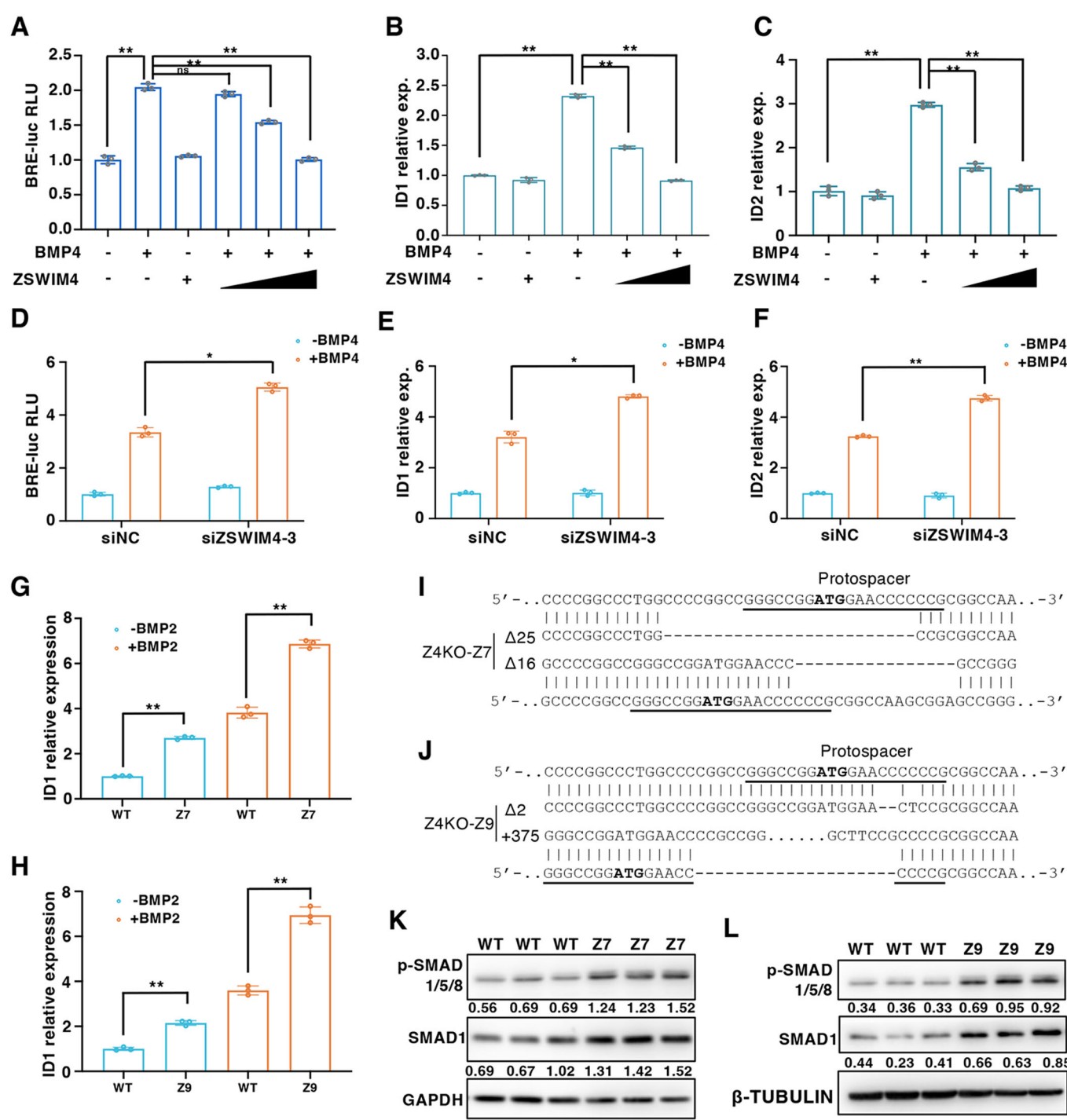

luciferase activity, and the expression of *ID1* and *ID2* (Fig. 4D–F). Moreover, we established *ZSWIM4* mutant HEK293T cell lines using CRISPR/Cas9. Two *ZSWIM4* mutant clones, Z7 (Δ25/Δ16) and Z9 (Δ2/+375), exhibited increased *ID1* expression with or without BMP2 treatment compared with wild-type cells (Fig. 4G±J). Consistent with this observation, the levels of p-SMAD1/5/8 and total SMAD1 were upregulated in the two mutant cell lines (Figs. 4K,L and EV5C,D). Collectively, these results indicate that the endogenous ZSWIM4 functions to attenuate the BMP signaling in HEK293T cells.

## ZSWIM4 promotes the ubiquitination and degradation of SMAD1

To examine the regulatory level of ZSWIM4 in the BMP signaling hierarchy, we transfected HEK293T cells with either constitutively active BMP receptor, *BMPR1A_CA* (Varley et al, 1998), or SMAD1 phosphomimetic, *SMAD1-DVD* (Nojima et al, 2010). Both components increased the BRE-luciferase activity, and this effect was blocked by the overexpression of *ZSWIM4* (Fig. 5A). This result suggests that ZSWIM4 acts downstream of SMAD1

**Figure 4. Zswim4 inhibits BMP signaling in HEK293T cells.**

(A–C) HEK293T cells were transfected with increasing doses of *ZSWIM4* (200, 300, and 400 ng/ml medium) and treated with BMP4 (100 ng/ml). Luciferase assays were performed to assess BMP signaling (A). The expression of *ID1* or *ID2* in the cells was examined by quantitative RT-PCR (B, C). β-*ACTIN* was used as the internal control. n = 3. (D–F) HEK293T cells were transfected with control or *ZSWIM4* siRNA (80 nM) and treated with BMP4 (100 ng/ml). The cells were collected for luciferase assays (D) or quantitative RT-PCR (E, F). n = 3. (G, H) Quantitative RT-PCR analysis of *ID1* expression in two HEK293T cell lines with *ZSWIM4* mutations with or without BMP2 treatment (100 ng/ml) treatment. n = 3. (I, J) DNA sequence alignments indicate the insertion or deletion in Z7 and Z9 cell lines with *ZSWIM4* mutation, respectively. sgRNA was designed to target the underlined sequence. (K, L) Western blot was performed to examine the change in p-SMAD1/5/8 and total SMAD1 in wild-type and *ZSWIM4* mutant cells. GAPDH or β-TUBULIN was used as the loading control. n = 3. Data information: n indicates biological replicates. Error bars show mean ± standard deviation (SD). Statistical analysis was performed using an unpaired Student's t test for (A–H). *p < 0.05, **p < 0.01, and ns indicates "not significant" (p > 0.05). Source data are available online for this figure.

phosphorylation, and its function is independent of the de-phosphorylation process of SMAD1, or at least, does not entirely rely on SMAD1 de-phosphorylation. Considering that the total SMAD1 protein levels were increased in the *ZSWIM4* mutant cell lines, Z7 and Z9, it is possible that ZSWIM4 attenuates BMP signaling by reducing SMAD1 protein stability. Indeed, both exogenous Myc-tagged *SMAD1* and endogenous SMAD1 proteins were reduced by increasing doses of *ZSWIM4* in a dose-dependent reduction (Appendix Fig. S3 and Fig. 5B). In *Xenopus* embryos injected with zswim4MO1, the endogenous Smad1 protein was upregulated in a dose-dependent manner compared with that in control embryos (Figs. 5C and EV5E). Thus, ZSWIM4 decreases SMAD1 protein levels in both *Xenopus* embryos and mammalian cells.

The mRNA levels of *SMAD1* were not affected by *ZSWIM4* overexpression in either *Xenopus* embryos or HEK293T cells (Appendix Fig. S4A,B). We then examined whether ZSWIM4 is involved in regulating SMAD1 protein stability. SMAD1 protein stability was measured following the treatment of HEK293T cells with cycloheximide (CHX) to inhibit protein synthesis. Indeed, the protein stability of SMAD1-FLAG was markedly reduced in the presence of ZSWIM4 compared with its stability in control cells (Figs. 5D,E and EV5F). We conclude that ZSWIM4 regulates the protein stability of SMAD1.

Given that ZSWIM4 is localized in the nucleus, we performed cell fractionation assays to determine whether only the nuclear SMAD1 was affected by ZSWIM4. After transfection of *ZSWIM4-Myc* and treatment with BMP4, we performed cell fractionation to separate cytoplasmic and nuclear proteins. A decrease in the total SMAD1 level was observed in the nuclear fraction after over-expression of *ZSWIM4*, compared with the levels in the nuclear fraction of control cells, while in the cytosol, the SMAD1 level remained at a similar level after overexpression of *ZSWIM4* (Figure EV5G,H). These results suggest that ZSWIM4 mainly reduced the total level of SMAD1 in the nuclear fraction, which is consistent with its nuclear localization.

The ubiquitination and proteasome-mediated degradation of SMAD1 represents an important regulatory mechanism in the BMP signal. To examine the ubiquitination of SMAD1, we co-transfected HA-tagged *SMAD1* and Myc-tagged *Ubiquitin* into HEK293T, and immunoprecipitation was performed. Western blots showed that the overexpression of *ZSWIM4* significantly increased the ubiquitination of SMAD1 (Figs. 5F and EV5I), while SMAD1 ubiquitination was reduced in the *ZSWIM4* mutant cell lines, Z7 or Z9 (Fig. 5G,H). Taken together, these results indicate that ZSWIM4 attenuates BMP signaling by promoting the ubiquitination and degradation of SMAD1.

The ubiquitin ligases SMURF1 and SMURF2 have been reported to promote the ubiquitination and degradation of SMAD1 (Zhang et al, 2001; Zhu et al, 1999). To examine whether ZSWIM4 is required for SMURF1 or SMRUF2 to regulate SMAD1 stability, we co-transfected *SMAD1-FLAG* and *SMURF1* or *SMURF2* into either the *ZSWIM4* mutant cell line, Z7, or wild-type HEK293T cells. The levels of SMAD1-FLAG were reduced by *SMURF1* or *SMURF2* overexpression in both cell lines (Figure EV5J), suggesting that ZSWIM4 is not involved in the degradation of SMAD1 mediated by SMURF1 or SMURF2.

## ZSWIM4 forms a complex with SMAD1 and the Elongin B/C-containing ubiquitin ligase

As ZSWIM4 regulates the ubiquitination of SMAD1, we next investigated whether it can physically interact with SMAD1. The FLAG-tagged *SMAD1* and Myc-tagged *ZSWIM4* were transfected into HEK293T cells, and co-immunoprecipitation (Co-IP) was performed with either anti-Myc or anti-FLAG antibodies. Strong interactions between SMAD1 and ZSWIM4 were observed in the co-transfected group (Fig. 5I). Co-IP was also performed using *Xenopus* embryos injected with *zswim4-FLAG* mRNA with anti-FLAG or anti-Smad1 antibodies. An interaction between Zswim4-FLAG and endogenous Smad1 was also detected in *Xenopus* embryos (Figure EV5K). The SMAD1 protein comprises three domains, i.e., the MH1 and MH2 domains, and a linker region. To identify the domain responsible for the interaction with ZSWIM4, we co-transfected HEK293T cells with *ZSWIM4-Myc* and different deletion mutants of *SMAD1* with a FLAG tag, respectively (Fig. 5J). After Co-IP, ZSWIM4-Myc was only co-precipitated with the SMAD1 deletion mutants containing an intact MH1 domain (Fig. 5K), indicating that the MH1 domain in SMAD1 is essential for its interaction with ZSWIM4. Accordingly, immunofluores-cence coupled with confocal microscopy revealed the co-localization of ZSWIM4 and SMAD1 in the nuclei of the HeLa cells (Figure EV5L). Thus, ZSWIM4 specifically interacts with SMAD1 in both *Xenopus* embryos and mammalian cell lines. In contrast, no interaction was detected between ZSWIM4 and SMAD2, the transducer protein of the Nodal signaling pathway (Appendix Fig. S5).

To identify the possible interaction partners of ZSWIM4 in regulating SMAD1 stability, stable isotope labeling by amino acid in cell culture (SILAC)-IP was performed (Fig. 6A; Pateetin et al, 2021; Zhang et al, 2022). After LC-MS/MS analysis of the immunoprecipitates, ELONGIN B (ELOB) and ELONGIN C (ELOC) were identified among the 10 most abundant proteins in forward and reverse SILAC (Fig. 6B). The interaction between

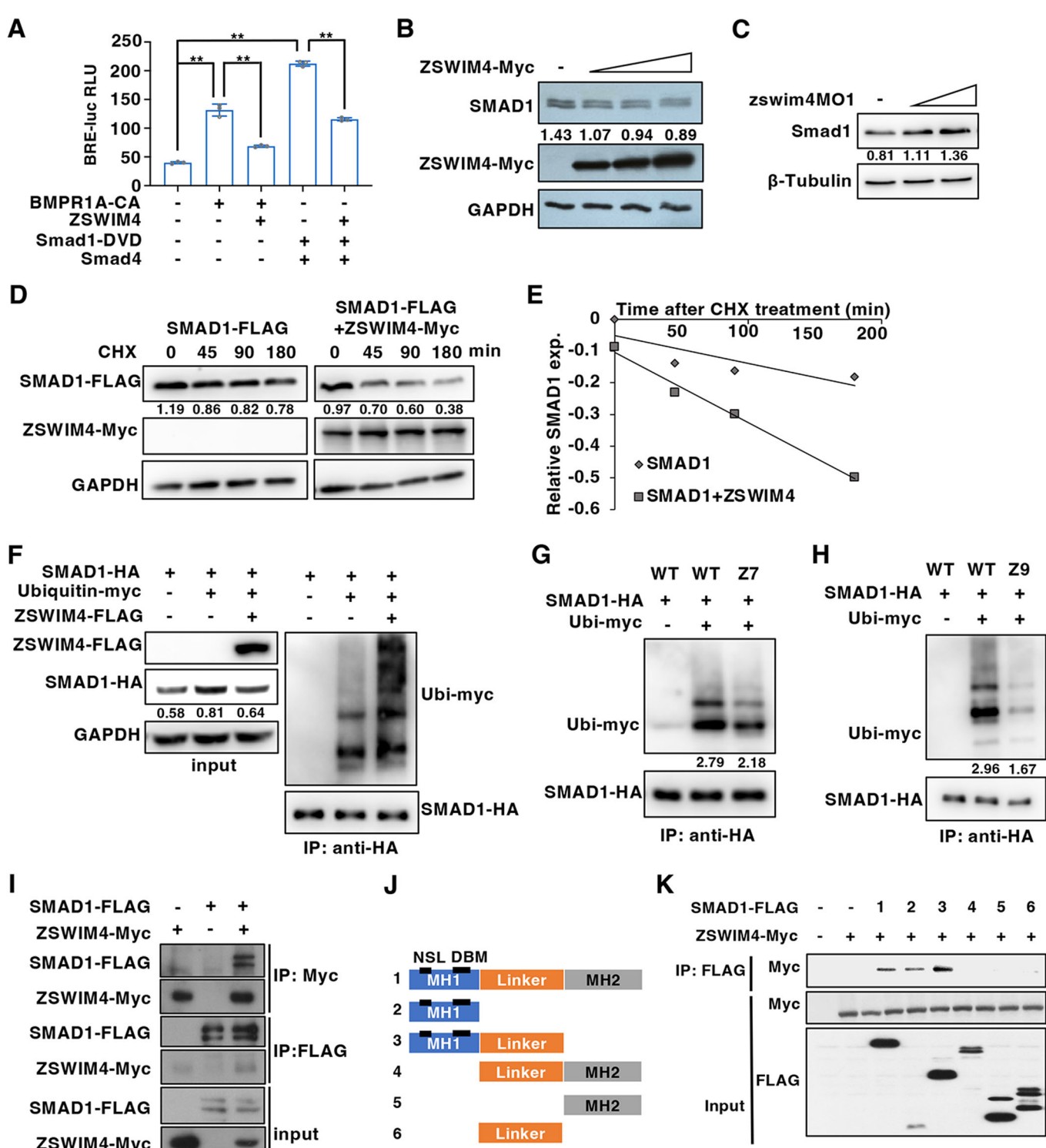

Zswim4 and ELOB or ELOC was then confirmed by Co-IP in HEK293T cells transfected with FLAG-tagged *zswim4* and Myc-tagged *ELOB* or *ELOC* (Fig. 6C,D). The ELOB and ELOC have been identified as positive regulators of RNA polymerase II (Bradsher et al, 1993; Bradsher et al, 1993). They also interact with Cullin and the RING domain protein Rbx to form the Cullin-RING ligase (CRL) complex, which promotes the ubiquitination of its targets by

binding to different substrate-recognition components (Petroski and Deshaies, 2005). ZSWIM4 was predicted to be a member of the Cul2-RING ubiquitin ligase complex using the online tool, STING. Previous studies showed that ZSWIM8 functions as a substrate-recognition subunit of the CRL complex (Han et al, 2020; Shi et al, 2020). We then performed an alignment using Zswim4 proteins from different species and human ZSWIM8, and a conserved

◄ **Figure 5.   ZSWIM4 interacts with SMAD1 and reduces the protein stability of SMAD1 by promoting its ubiquitination.**

(A) The BRE-luciferase assay was performed in HEK293T cells transfected with either constitutively active BMP receptor (*BMPR1A_CA*), Smad1 phosphomimetic construct (*Smad1-DVD*) alone, or the combination with *ZSWIM4* (300 ng/ml medium) as indicated. $n = 3$. (B, C) Western blot analysis of endogenous SMAD1 protein levels in HEK293T cells transfected with increasing doses of *ZSWIM4-Myc* (200, 300, and 400 ng/ml medium) (B), or in *X. tropicalis* embryos of stage 11 injected with zswim4MO1 (4 ng/embryo or 8 ng/embryo). $n = 2$. (D, E) *SMAD1-FLAG* was transfected either alone or together with *ZSWIM4-Myc* (300 ng/ml medium) into HEK293T cells. After 24 h, the cells were treated with cycloheximide (CHX, 50 μg/ml) for indicated periods, and the protein level of SMAD1-FLAG was examined by western blot. The relative protein level of SMAD1-FLAG was quantified by Image J and divided by GAPDH, which was further normalized to that at 0 min. $n = 2$. (F–H) HA-tagged *SMAD1* and Myc-tagged *Ubiquitin* were co-transfected into Z7 (G), Z9 (H), or wild-type HEK293T cells (F) with *ZSWIM4* overexpression (300 ng/ml medium). Cell lysates were collected for immunoprecipitation using anti-HA antibody. Ubiquitination of SMAD1-HA was detected by western blot using anti-Myc antibody. $n = 2$. (I) Co-immunoprecipitation (Co-IP) was performed in HEK293T cells transfected with either *SMAD1-FLAG*, *ZSWIM4-Myc* alone, or the combination of both. $n = 2$. One representative blot is shown. (J, K) A schematic diagram of SMAD1 deletion mutants and their binding to ZSWIM4-Myc was examined using Co-IP. For the western blot in (B–D, F–I, K), $n = 2$. One representative blot is shown. Data information: *n* indicates biological replicates. Error bars show mean ± standard deviation (SD). Statistical analysis was performed using an unpaired Student's *t* test for (A). **$p < 0.01$. Source data are available online for this figure.

BC-box and Cul2-box were identified in the N-terminus of Zswim4 (Fig. 6E; Mahrour et al, 2008). Indeed, we found that Zswim4 has a physical interaction with CUL2, the scaffold protein in the CRL complex (Fig. 6F). To examine whether ZSWIM4 interacts with ELOB or ELOC through the BC-box, we generated a ZSWIM4 deletion mutation, ΔN, by removing the N-terminal amino acids 1–99, which encompass the BC-box (Fig. 6G). Indeed, ΔN exhibited a substantially diminished binding ability to ELOB and ELOC than did wild-type ZSWIM4 (Fig. 6H,I). Thus, ZSWIM4 interacts with the components of the CRL complex, and the amino acids 1-99 are essential for binding to ELOB and ELOC. Taken together, these results indicate that ZSWIM4 forms a complex with CRL and SMAD1.

## ZSWIM4 functions in the CRL complex to regulate SMAD1 stability

As ZSWIM4 interacts with SMAD1 and CRL, it may function as a substrate recognition unit for the CRL complex. To examine whether ZSWIM4 regulates SMAD1 protein degradation by cooperating with CRL, we co-expressed ZSWIM4 and the components of the CRL complex. Compared with the single transfection of *ZSWIM4-Myc* or *cul2*, co-transfection of both showed a stronger effect in reducing the stability of Myc-tagged SMAD1 (Appendix Fig. S6). Further co-transfection of *ELOB* and *ELOC* caused the highest degree of degradation of SMAD1-Myc (Figs. 7A and EV5M). Consistent with these results, the ubiquitination assay of SMAD1-HA showed the strongest signals in the cells co-overexpressing Zswim4-FLAG, Cul2, ELOB, and ELOC (Figs. 7B and EV5N). Furthermore, luciferase assays were performed to examine the effects of these factors on BMP signaling activity. Transfection of a low dose of *zswim4* or *cul2* had little effect on the luciferase activity, while their co-transfection apparently reduced the luciferase activity (Fig. 7C). Luciferase activity was further reduced when *ELOB* and *ELOC* were also transfected, either in the presence or absence of BMP2 (Fig. 7C). These results showed that ZSWIM4 cooperates with the components of CRL complex to regulate BMP signaling in HEK293T cell line. To determine if a similar interaction occurs in *Xenopus* embryos, we first examined the spatial expression patterns of *cul2*, *elob* and *eloc*. Similar to *zswim4*, *cul2*, *elob* and *eloc* showed strong expression in animal pole region at cleavage and blastula stages (Appendix Fig. S7). At gastrula stage, *elob* and *cul2* were also expressed in the dorsal blastopore lip and dorsal ectoderm (Appendix Fig. S7). Next, an animal cap assay was performed to

study whether the CRL complex can attenuate the BMP signal in *Xenopus* embryos. *bmp4* mRNA was co-injected with *cul2* or human *ELOB* and *ELOC* mRNAs. Overexpression of *cul2* or *ELOB* and *ELOC* efficiently attenuated the *bmp4*-induced expression of *epiker*, *msx1* and *xbra*, (Fig. 7D), suggesting a conserved role of CRL in attenuating BMP signaling in *Xenopus* animal caps. It is notable that overexpression of *cul2* alone or combination of *ELOB* and *ELOC* did not reduce the expression of *msx1* and *epiker* in animal caps without BMP activation (Fig. 7D). This suggests that their inhibitory activity against BMP is reliant on additional factors or the presence of an external BMP activating environment. In subsequent animal cap assays, co-injection with low doses of *zswim4*, *cul2* and *ELOB/ELOC* significantly induced neural markers, *sox2* and *sox3* (Fig. 7E). This induction was not observed when these components were injected individually, suggesting a synergistic effect of Zswim4 and Cul2, ELOB/ELOC in neural induction (Fig. 7E). Additionally, we performed luciferase assays in whole embryos injected with BRE-luciferase reporter with low doses of *zswim4*, *cul2* and *ELOB/ELOC*, specifically targeting the two dorsal blastomeres at the four-cell stage. Compared with the individual components, co-injection of *zswim4*, *cul2* and *ELOB/ELOC* caused a more pronounced reduction of luciferase activity that was induced by Bmp4 (Fig. 7F). These findings suggest that Zswim4, Cul2 and ELOB/ELOC cooperate to repress BMP signaling in *Xenopus* embryo. Taken all together, our data illuminate a novel interaction where ZSWIM4 and the CRL complex work synergistically to attenuate SMAD1 protein stability, thereby attenuating BMP signaling.

## Discussion

Zswim4 is a conserved protein, but its functions are largely unknown. In this study, we investigated the function of Zswim4 during early embryonic development and elucidated its function in mediating BMP signaling transduction. Our results indicate that it attenuates BMP signaling and is essential for axis formation during gastrulation. Indeed, ZSWIM4 is a novel BMP inhibitor that is involved in the SMAD1 degradation in the nucleus.

Zswim4 is expressed in the SMO and then enriched in the dorsal ectoderm at the mid-gastrula stages. The SMO plays an essential role during the neural induction, the development of the anterioposterior (Jansen et al, 2007) and dorsoventral pattern (Kiecker and Lumsden, 2012). Our observations on Zswim4 are consistent with these functions. The gain-of-function of *zswim4* led

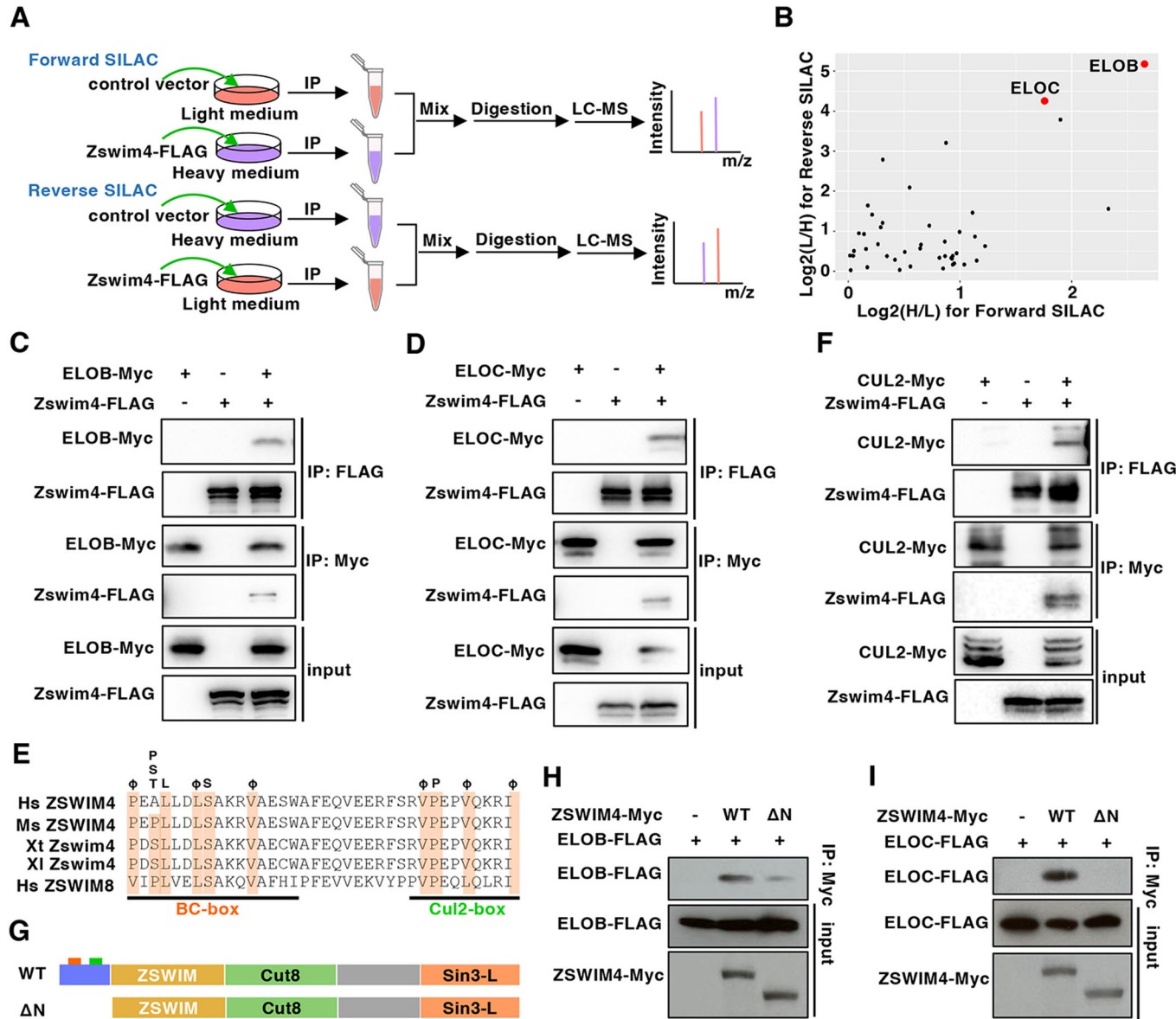

**Figure 6.   ZSWIM4 interacts with components of the CRL complex.**

(A) SILAC-based quantitative mass spectrometry was used to identify the interaction partners of Zswim4 in the HEK293T cells. (B) ELOB and ELOC rank high among the common proteins identified in both forward and reverse SILAC assays. (C, D) Co-IP was performed to examine the interaction between FLAG-tagged Zswim4 and Myc-tagged ELOB, ELOC. $n = 2$. One representative blot is shown. (E) Sequence alignment illustrates the conserved N-terminus (amino acids 32-67 in human ZSWIM4) among human ZSWIM8 and Zswim4 from different species. Hs, *Homo sapiens*; Ms, *Mus musculus*; Xt, *Xenopus tropicalis*; Xl, *Xenopus laevis*. (F) The physical interaction between Zswim4 and CUL2 is confirmed by Co-IP. $n = 2$. One representative blot is shown. (G–I) Co-IP results showed that the N-terminal region of ZSWIM4 is responsible for its interaction with ELOB (H) or ELOC (I). ΔN represents the deletion mutant of ZSWIM4 with N-terminal 99 amino acids with the BC-box deleted (G). $n = 2$. One representative blot is shown. Data information: *n* indicates biological replicates. Source data are available online for this figure.

to a shortened anteroposterior axis. Knockdown of *zswim4* in *X. tropicalis* embryos resulted in ventralization, which was also observed in *zswim4* mutant F1 generation embryos (Fig. 2H). The SMO attenuates the Wnt, BMP, and Nodal signaling during gastrulation by secreting their antagonists. Chordin and Noggin are the two secreted BMP antagonists that help establish a BMP gradient along the dorsoventral axis by blocking the binding of BMP ligands to their receptors (Inomata et al, 2013). In contrast to the secreted antagonists that work extracellularly, ZSWIM4

attenuates BMP signaling by inducing the degradation of nuclear SMAD1 in the nucleus. The phosphorylation of Smad1, as an indicator of BMP signaling activation, was also downregulated by Zswim4 (Fig. 3S), which may be mainly due to the degradation of the nuclear Smad1. The BRE luciferase activity induced by a constitutively active BMP receptor or by phosphomimetic Smad1-DVD was attenuated by *ZSWIM4* overexpression to nearly the same extent (Fig. 5A), suggesting that direct de-phosphorylation of Smad1 by Zswim4 plays a minor role, if any, in this process. Thus,

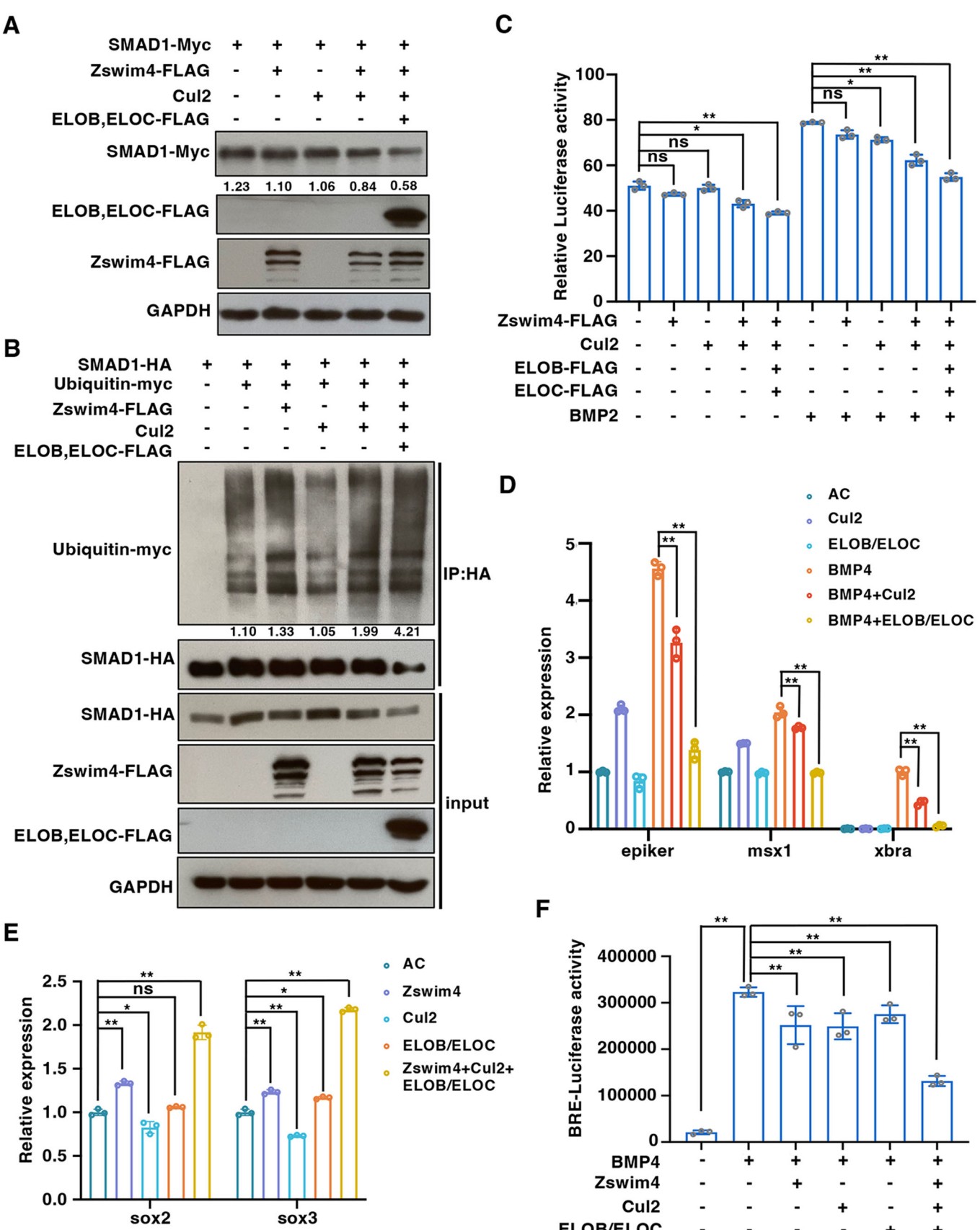

**Figure 7.  Zswim4 cooperates with the CRL complex to regulate BMP signaling.**

(**A**) Western blot analysis of SMAD1-Myc protein levels in HEK293T cells transfected with *zswim4-FLAG* (150 ng/ml medium), *cul2* (150 ng/ml medium), *ELOB-FLAG* (100 ng/ml medium), and *ELOC-FLAG* (100 ng/ml medium). *n* = 2. (**B**) SMAD-HA ubiquitination was examined in HEK293T cells co-transfected with either *zswim4-FLAG* (150 ng/ml medium), *cul2* (150 ng/ml medium) alone or together with *ELOB-FLAG* (100 ng/ml medium) and *ELOC-FLAG* (100 ng/ml medium). The intensities of the SMAD1-HA ubiquitination bands were normalized to the responding immunoprecipitated SMAD1-HA bands. *n* = 2. (**C**) Luciferase assay was performed on HEK293T cells transfected with *zswim4* and the CRL components as indicated (150 ng *zswim4-FLAG*/ml medium, 150 ng *cul2*/ml medium, 100 ng *ELOB-FLAG* /ml medium, and 100 ng *ELOC-FLAG*/ml medium) with or without BMP2 treatment (100 ng/μl). *n* = 3. (**D, E**) *X. laevis* embryos were injected with mRNAs as indicated (300 pg *bmp* mRNA/ embryo, 200 pg *zswim4* mRNA/embryo, 200 pg *cul2* mRNA/embryo, 100 pg *ELOB* mRNA/embryo, 100 pg *ELOC* mRNA/embryo). Animal caps were dissected at stage 9 and cultured for 2 h. Quantitative RT-PCR was used to examine the expression of the BMP target genes, *epiker*, *msx1* and *xbra* (**D**) or neural markers, *sox2* and *sox3* (**E**). *n* = 3. (**F**) Luciferase assay was performed using whole *X. laevis* embryos injected with indicated mRNAs to the two dorsal blastomeres at the four-cell stage. For the western blot in (**A, B**), *n* = 2, biological replicates. One representative blot is shown. *n* = 3. Data information: *n* indicates biological replicates. Error bars show mean ± standard deviation (SD). Statistical analysis was performed using an unpaired Student's *t* test for (**C–F**). **p* < 0.05, ***p* < 0.01, and ns indicates "not significant" (*p* > 0.05). Source data are available online for this figure.

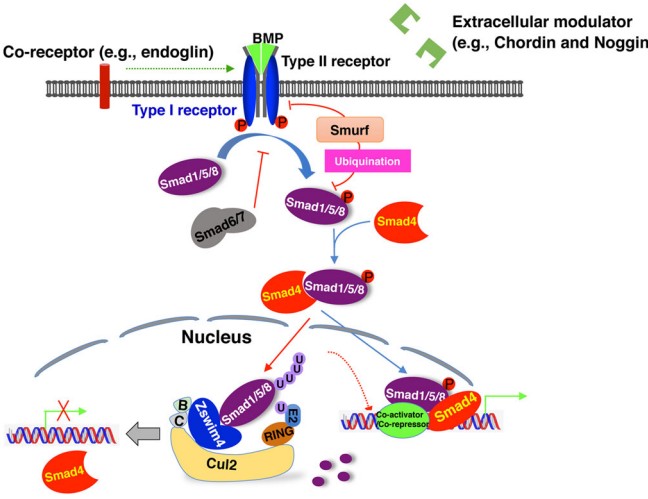

**Figure 8.   A schematic diagram illustrating the function of ZSWIM4 in BMP signaling.**

The binding of BMP ligands to their receptors induces phosphorylation of R-Smads (Smad1/5/8). The phosphorylated R-Smads interact with Smad4 to form a complex that translocates into the nucleus and activates the target genes. Zswim4 interacts with Cullin2 and Elongin B and C, forming a protein disruption complex to promote Smad1 ubiquitination and degradation in the nucleus.

in addition to the secreting antagonists and inhibitory SMADs, we identified a mechanism by which ZSWIM4 regulates BMP signaling in the nucleus.

Ubiquitin-mediated proteasomal degradation of SMAD1 represents an important mechanism to regulate BMP signaling. Smurf1 and Smurf2 were reported to induce the ubiquitination and degradation of R-Smads in BMP signaling pathway (Zhang et al, 2001; Zhu et al, 1999). The SMURF1/2 mediated SMAD1 ubiquitination was maintained at a similar level in *ZSWIM4* mutant cells to that in WT HEK293T cells, suggesting ZSWIM4 regulates SMAD1 stability by another mechanism. Indeed, ELOB and ELOC, the two components of the CRL complex, were identified as ZSWIM4 interaction partners by SILAC assay. We also confirm the physical interaction between the ZSWIM4 and CUL2. CRL complexes utilize the CUL2 as the scaffold and recruit diverse substrate-receptors to target different proteins (Petroski and Deshaies, 2005). ZSWIM4 interacts with SMAD1 and the CRL

components, CUL2, ELOB and ELOC, and co-expression of them synergistically promotes SMAD1 ubiquitination and degradation. These observations support the model that ZSWIM4 works as a substrate-recognition subunit of CRL to regulate SMAD1 stability (Fig. 8). Confocal imaging showed that ZSWIM4 proteins were localized in the nucleus in both *Xenopus* embryos and Hela cell line (Figs. 1E,F and EV1M), which implies that ZSWIM4 functions in the nucleus to regulate SMAD1 degradation. In line with the model, a decrease in the total SMAD1 level was only observed in the nuclear portion after overexpression of *ZSWIM4*, but not in the cytosol (Fig EV5G,H). Zswim4 and Smurf1/2 represent two independent mechanisms in regulating Smad1 degradation and BMP signaling pathway during *Xenopus* early gastrulation. Injection with *zswim4* into the ventral region of the *Xenopus* embryo did not induce the secondary axis. This observation suggests that Zswim4 serves as a fine-tuned modulator of BMP signaling. It is likely that Zswim4 acts in coordination with other strong regulators to achieve precise and accurate regulation of BMP signaling during embryonic patterning.

The Zswim family consists of at least nine members. To date, only a few studies have reported the functions of the family members. ZSWIM1 is a novel biomarker in T helper cell differentiation (Ko et al, 2014). A mutation in ZSWIM6 causes acromelic frontonasal dysostosis (AFND) which is characterized by craniofacial, brain, and limb malformations, perhaps by inhibiting Hedgehog signaling (Twigg et al, 2016). Zswim6 also contains a Cul2-box (Tischfield et al, 2017), and may function as an E3 ubiquitin ligase. In *C. elegans*, the Zswim8 homolog, EBAX-1, regulates axon guidance by functioning as a substrate-recognition subunit in the CRL complex (Wang et al, 2013). Two recent studies also showed that ZSWIM8 is involved in target-directed miRNA degradation in cells of different species by functioning in the CRL complex to degrade Argonaute proteins, with Zswim8 acting as a substrate receptor (Han et al, 2020; Shi et al, 2020). In our study, ZSWIM4 was found to cooperate with the CRL complex to regulate the ubiquitination and degradation of SMAD1 protein. Moreover, ZSWIM4 was found to regulate BMP signaling and early embryonic development in *Xenopus* embryos. Although Zswim family members are involved in various biological processes, studies by us and others suggest that they function, at least in part, as substrate adapters for the CRL complex using similar molecular mechanisms. This may explain the strong genetic compensation effect observed in our *X. tropicalis* zswim4F2 and *MZzswim4*-/- embryos, in which other Zswim family members were upregulated

(Figure EV4A), and may have compensated for Zswim4 function in the *zswim4* mutant embryos. Our results confirmed the contribution of *zswim7* to the genetic compensation in which overexpression of *zswim7* has a rescue effect on *zswim4* crispants and knockdown of *zswim7* in *zswim4F2* increased the ratio of defective embryos (Figure EV4B–G). Taken together, we identified ZSWIM4 as a novel BMP inhibitor that promotes SMAD1 ubiquitination in the nucleus, and plays an essential role during embryonic body axis formation.

## Methods

### Plasmid constructs

The open reading frames of human and *Xenopus zswim4, cul2*, and human *ELOB, ELOC*, and *SMAD1* were amplified by PCR and subcloned into pCS2$^+$ or pCS2$^+$-MT, pCS2$^+$-FLAG, pCS2$^+$-HA vectors. Truncated mutants of human *ZSWIM4* were generated by PCR and subcloned into pCS2$^+$-MT. All the constructed plasmids were confirmed by Sanger sequencing. pSpCas9(BB)-2A-GFP (PX458) was purchased from Addgene (Addgene plasmid # 48138), and was used to generate *ZSWIM4* mutant cell line (Ran et al, 2013). pCMV5-hBMPR-1A CA (# 49527) and DR274 (# 42250) were also obtained from Addgene.

### Cell culture and transfection

HEK293T and Hela cells were cultured in Dulbecco's modified Eagle's medium with 10% fetal bovine serum in a culture incubator at 37 °C under 5% CO$_2$. Transfection was performed using Lipofectamine 2000 (Invitrogen), following the manufacturer's instructions.

### Embryo manipulations and injection

*X. laevis* and *X. tropicalis* juveniles or adult frogs were purchased from Xenopus 1 Company (USA), and they were raised in a circulating aquatic system. The use of frogs for this study was approved by the Ethics Committee of The Chinese University of Hong Kong and licensed by the Department of Health, Hong Kong. *X. laevis* embryos were obtained by in vitro fertilization (Shi et al, 2015), and *X. tropicalis* embryos were obtained by natural mating. To prime the frogs, the male and female *X. tropicalis* were injected with 20 units of hCG into the dorsal lymph sac. On the next day, 200 units of hCG were injected for induction of full ovulation. Embryos were staged according to Nieuwkoop and Faber (Nieuwkoop and Faber, 1967). MOs targeting *zswim4* and *zswim7* were purchased from Gene Tools, Inc. (Appendix Table S1). Capped mRNAs and sgRNAs were in vitro transcribed as previously described (Wang et al, 2020). mRNAs or MOs were injected at the two-cell or four-cell stage. For CRISPR/Cas9-mediated gene disruption, the sgRNAs were co-injected with CAS9 protein (PNA Bio) into the one-cell stage *X. tropicalis* embryos. Quantification of phenotypes in injected embryo was perform by two researchers to minimize the effects of subjective bias, and no blinding was done.

### Whole-mount in situ hybridization and sectioning

*Xenopus* embryos were collected at the indicated stage, fixed in HEMFA (Wang et al, 2021), and whole-mount in situ hybridizations were performed as previously described (Harland, 1991; Wang et al, 2019). After in situ hybridization, the embryos were embedded and sectioned according to the protocol used in our previous study (Wang et al, 2011).

### Animal cap assay, RT-PCR, and real-time PCR

For animal cap assay, indicated mRNAs or MOs were injected into the animal pole of *X. laevis* embryos at the two-cell stage. When the embryos developed to stage 9, the animal caps were dissected, and incubated in the L-15 medium for 2 h (Zhao et al, 2008). RNA was then extracted using TRIzol reagent, and reverse transcriptions were performed using Superscript III (Thermo Fisher Scientific). Real-time PCR was performed using a ViiA 7 Real-Time PCR system (Applied Biosystems) using TB Green PCR premix (Takara). The Primers used for semi-quantitative RT-PCR and quantitative RT-PCR are listed in Appendix Table S2.

### Luciferase assay

HEK293T cells were seeded into a 24-well plate and transfected with a BRE reporter plasmid, Renilla luciferase pRL, and the indicated plasmids. One day after transfection, the cells were serum-starved for 2 h, and then BMP2 or BMP4 was added. Luciferase activity was measured 18 h later using the Dual-Luciferase Assay system (Promega).

### Co-immunoprecipitation (Co-IP)

Co-IP was performed as previously described (Wang et al, 2015). Briefly, the lysates of *Xenopus* embryos or HEK293T cells were incubated with the indicated antibody at 4 °C overnight, followed by incubation with protein G Dynabeads (Thermo Fisher Scientific) for 1 h and then washing in lysis buffer five times. The precipitants and whole cell lysates were separated by SDS-PAGE and transferred to the nitrocellulose membrane for blotting.

### Immunofluorescence

Cells were plated on poly-lysine coated glass slides. Twenty-four hours after transfection, the glass slides were washed twice with cold 1× PBS. The cells adhered on the glass slides were fixed with 2% PFA in 1× PBS and then incubated with the desired antibodies for 1 h. Alexa Fluor 488- or 568-conjugated secondary antibodies were used to stain the first antibody. Fluorescence signals were captured by using an Olympus FV3000 confocal microscope.

### Stable Isotope Labeling by Amino acid in Cell culture (SILAC)-IP for ZSWIM4 interactomics

Samples were prepared for SILAC using the SILAC Protein Quantitation Kit (Thermo Fisher Scientific #A33972) according to the manufacturer's protocol. Briefly, HEK293T cells were cultured in DMEM, and 10% dialyzed fetal bovine serum, with L-Arginine monohydrochloride and L-Lysine monohydrate (Light amino acids) or $^{15}N_2$$^{13}C_6$-lysine and $^{15}N_4$$^{13}C_6$-arginine (Heavy amino acids). After eight passages, cells were collected from the heavy samples to verify the efficiency of isotope incorporation by

mass spectrometry. For forward SILAC assays, the Zswim4-FLAG expression construct was transfected into the heavy sample, and the construct with a FLAG tag only was transfected into the light sample. For reverse SILAC assays, the Zswim4-FLAG construct was transfected into the light sample. IP was then performed with anti-FLAG antibodies as in the Co-IP protocol described above. The eluted precipitants were quantified and digested with trypsin, then desalted and dried in a vacuum evaporator. The precipitants from light and heavy samples were combined in a 1:1 ratio for LC-MS/MS analysis.

## Proteins, antibodies, and siRNAs

Recombinant BMP2 or BMP4 proteins were purchased from Cell Signaling Technology (CST). The antibodies used in this study are listed in Appendix Table S3. Small interfering RNAs (siRNA) targeting *ZSWIM4* were purchased from Genepharma (Shanghai, China). The sequences of the synthetic siRNAs are siRNA1 5′GCGUGUUGCAA GUGGGAUUTT3′, siRNA2 5′GGAGGAGACACUUACCCUUTT3′, and siRNA3 5′GCCCGCAAGCCCUGAUGAATT3′.

# Data availability

Mass spectrometry proteomics data have been submitted to the ProteomeXchange Consortium: PXD035207.

# Peer review information

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

## Acknowledgements

This work was supported by the the grants from the National Key R&D Program of China, Synthetic Biology Research (2019YFA0904500), the

Research Grants Council of Hong Kong (14112618 and 14119120) to HZ, and CRF equipment grant C5033-19E. Additional support was provided by the Hong Kong Branch of CAS Center for Excellence in Animal Evolution and Genetics to HZ (8601012). We thank our laboratory colleagues and the colleagues in core lab of School of Biomedical Sciences, CUHK, for their helpful discussion in this project.

## Author contributions

**Chengdong Wang**: Data curation; Formal analysis; Investigation; Methodology; Writing—original draft. **Ziran Liu**: Data curation; Formal analysis; Methodology; Writing—original draft. **Yelin Zeng**: Data curation; Formal analysis; Methodology. **Liangji Zhou**: Data curation; Formal analysis; Methodology. **Qi Long**: Data curation; Formal analysis; Methodology. **Imtiaz Ul Hassan**: Data curation; Formal analysis; Methodology. **Yuanliang Zhang**: Software; Formal analysis. **Xufeng Qi**: Resources; Writing—review and editing. **Dongqing Cai**: Resources; Writing—review and editing. **Bingyu Mao**: Resources. **Gang Lu**: Writing—review and editing. **Jianmin Sun**: Writing—review and editing. **Yonggang Yao**: Writing—review and editing. **Yi Deng**: Resources; Writing—review and editing. **Qian Zhao**: Software. **Bo Feng**: Writing—review and editing. **Qin Zhou**: Writing—review and editing. **Wai Yee Chan**: Writing—review and editing. **Hui Zhao**: Conceptualization; Resources; Formal analysis; Supervision; Funding acquisition; Investigation; Project administration; Writing—review and editing.

## Disclosure and competing interests statement

The authors declare no competing interests.

# Expanded View Figures

**Figure EV1.  Characterization of *Xenopus zswim4*.**                                                                                      ▶

(**A**, **B**) Zswim4 is highly conservative among different species. Sequence alignment (**A**) and sequence identities (**B**) among *X. laevis*, *X. tropicalis*, mouse, and human ZSWIM4. (**C–L**) Spatial expression pattern of *zswim4* in *Xenopus* embryos revealed by whole mount in situ hybridization. (**C**, **D**) animal pole view; (**E**) dorsal vegetal view; (**F**) lateral view with dorsal towards right; (**G**, **H**) dorsal view; (**I**, **J**) lateral view; (**K**, **L**) Transverse section of a stage 28 embryo at the levels illustrated by black lines in (**I**). nt neural tube, e eye, fg foregut, nc notochord. Scale bar = 250 μm. (**M**) Immunofluorescence staining of *X. laevis* embryos at stage 10 using anti-Rabbit IgG or anti-Zswim4 antibody, with dorsal side towards right. Scale bar = 200 μm. Source data are available online for this figure.

                                                                 

**A**

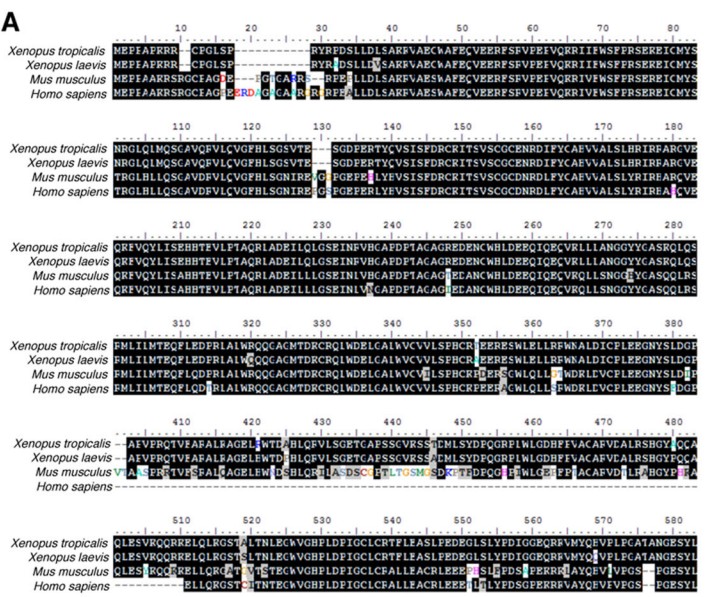

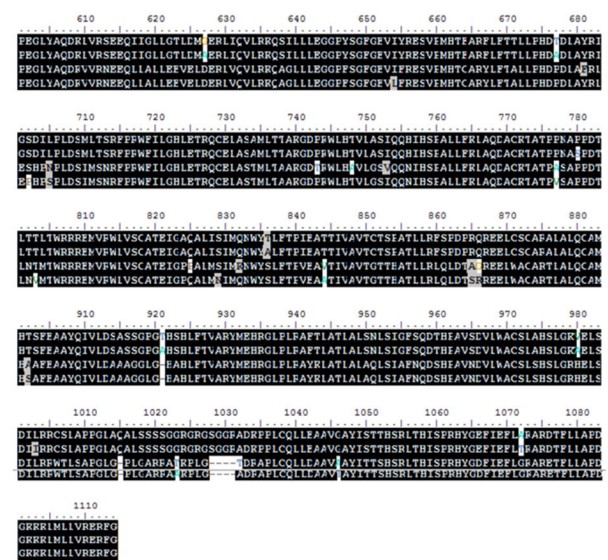

**B**

| Species | Identity |
|---------|----------|
| *Xenupus laevis* | --- |
| *Xenopus tropicalis* | 98.1% |
| *Mus musculus* | 74.3% |
| *Homo sapiens* | 69.9% |

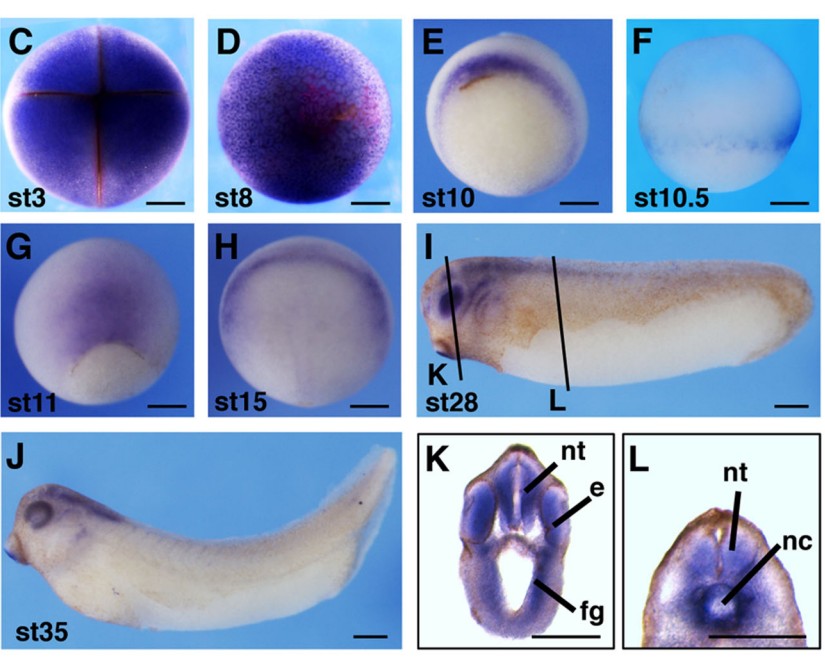

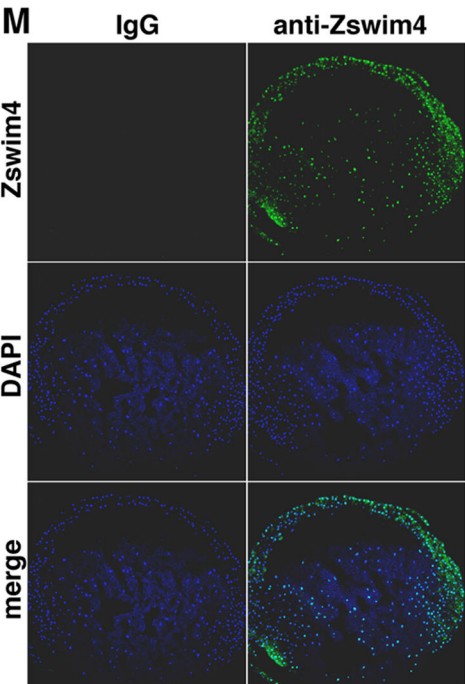

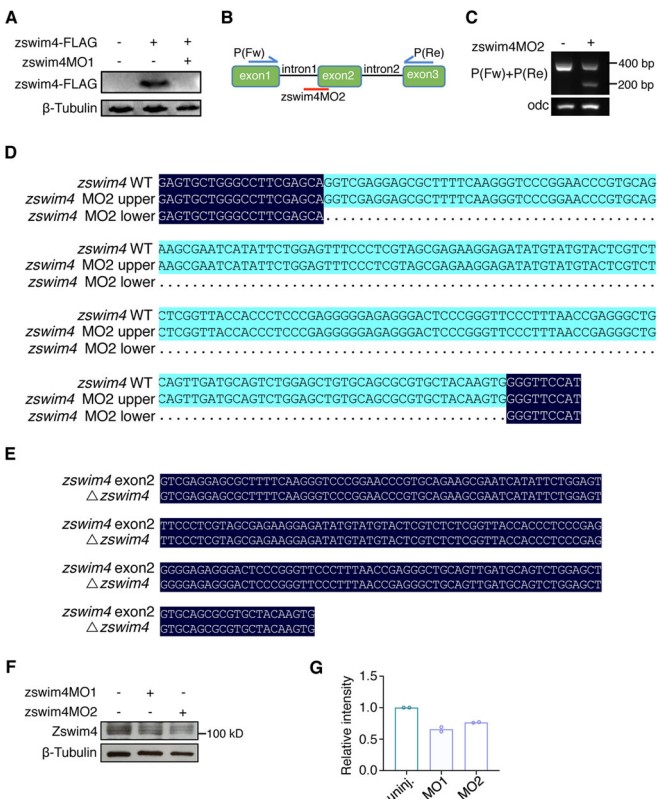

**Figure EV2.   Validation of zswim4MO1 and zswim4MO2.**

(**A**) Western blot analysis of *zswim4-FLAG* protein in *X. tropicalis* embryos injected with *zswim4-FLAG* mRNA (60 pg/embryo) and zswim4MO1 (6 ng/embryo). β-Tubulin was used as a loading control. $n = 2$. One representative blot is shown. (**B–E**) Schematic diagram showing the zswim4MO2 binding site and the primer pair used to amplify the exon2 (**B**). RT-PCR was performed using the primer pair covering the *zswim4* exon2, and two bands were amplified from the *X. tropicalis* embryos injected with zswim4MO2 (**C**). The two bands were recovered for Sanger sequencing. The sequencing results were aligned with wild-type *zswim4* (**D**). The lost sequence in the lower band has 100% identity with *zswim4* exon2 (**E**). For western blot in (**A**) and RT-PCR in (**C**), $n = 2$. One representative blot is shown. (**F, G**) Western blot analysis of endogenous Zswim4 protein in *X. tropicalis* embryos injected with 6 ng of zswim4MO1 or zswim4MO2. β-Tubulin was used as a loading control. Quantification of Zswim4 bands is shown in (**G**). $n = 2$. Data information: *n* indicates biological replicates. Source data are available online for this figure.

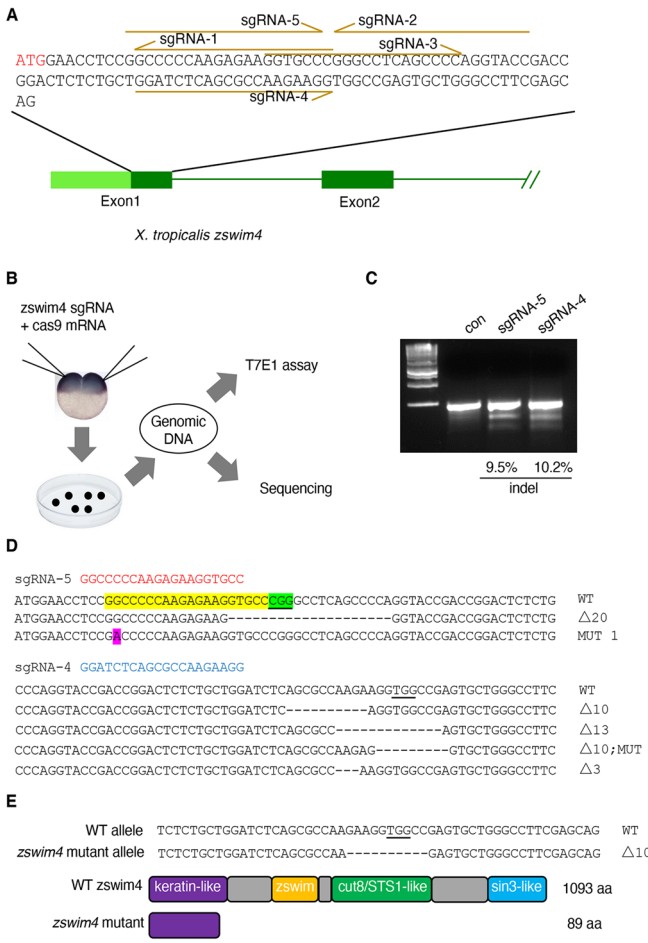

**A** Five sgRNAs were designed to target *X. tropicalis zswim4* exon1. The start codon is marked in red. (**B**) Schematic diagram indicating the generation of *zswim4* mutant embryos using CRISPR/Cas9. (**C**) T7E1 assay was performed using genomic DNA extracts from the injected or uninjected embryos. $n = 2$. One representative blot is shown. (**D**) Sanger sequencing results confirmed the mutations in the *zswim4* CRISPR target region. (**E**) The *zswim4* mutant allele with a 10-bp deletion and the putative peptide encoded by the *zswim4* Δ10 mutant. Data information: *n* indicates biological replicates. Source data are available online for this figure.

**Figure EV3. Generation of the *zswim4* mutant frog line using CRISPR/Cas9.**

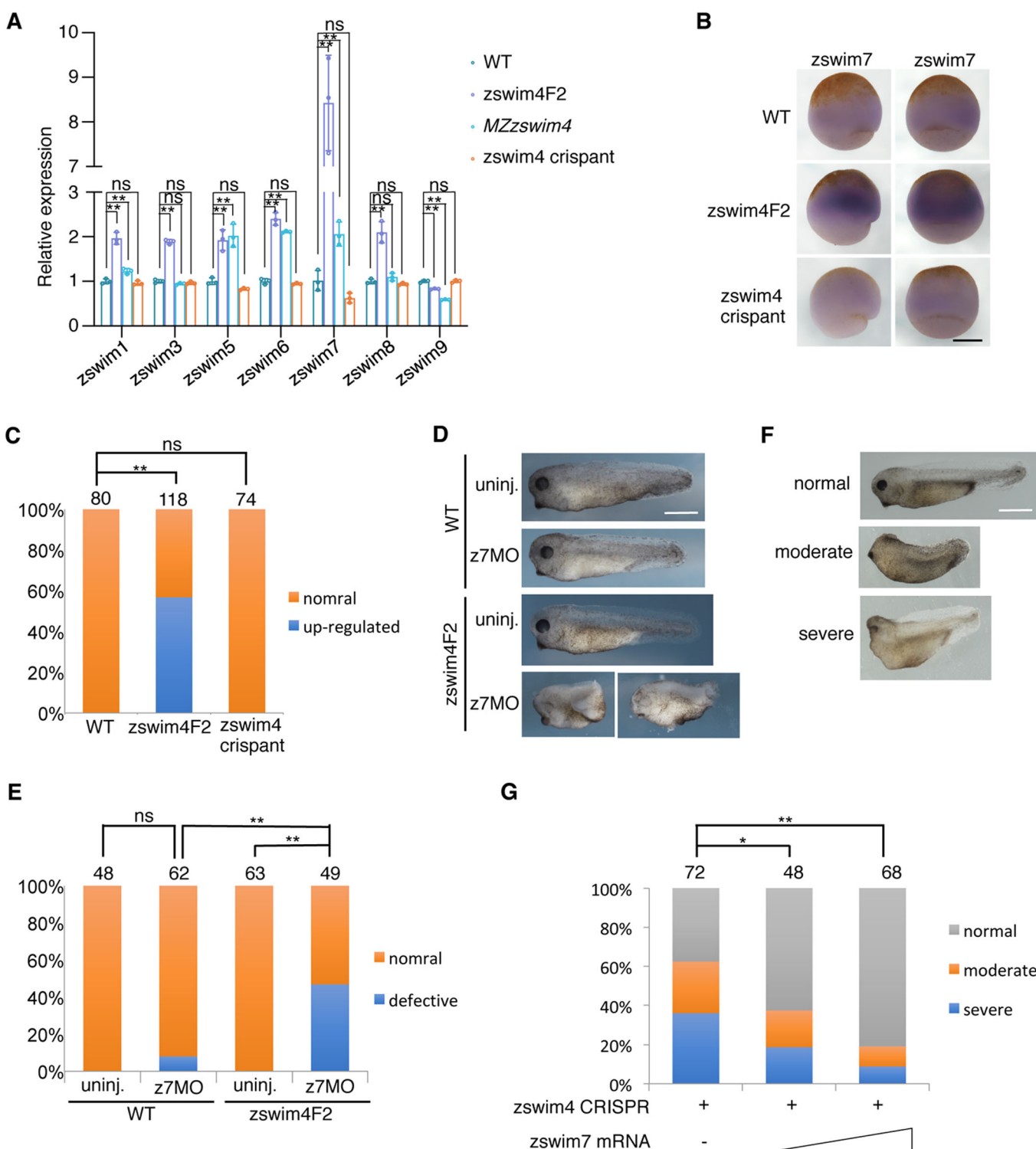

**Figure EV4.  Up-regulated *zswim7* contributes to the genetic compensation in *zswim4* mutant frog lines.**

(A) Genetic compensation was observed in *zswim4F2* embryos or *MZzswim4$^{-/-}$* embryos. After crossing two *zswim4$^{+/-}$* or two *zswim4$^{-/-}$*, the offsprings of stage 10 were collected for RNA extraction. The F0 embryos injected with Cas9 protein and four sgRNAs were also collected. Quantitative RT-PCR was performed to determine the gene expression of the indicated *zswim* family members. *odc* was used as the internal standard. $n = 3$. (B, C) WMISH showing the expression of *zswim7* in *X. tropicalis* embryos, *zswim4F2* embryos or zswim4 crispants (injected with 200 pg sgRNA plus 0.5 ng Cas9 protein/embryo). Embryos in the left column, lateral view with the dorsal side towards the right. Embryos in the right column, dorsal view. Scale bar = 1000 μm. The ratio of embryos with up-regulated expression of *zswim7* is shown in (C). The numbers on the top indicate the total number of embryos. $n = 3$. (D, E) *zswim4F2* embryos were injected with 6 ng of morpholino targeting *zswim7* (z7MO) at the one- or two-cell stage, and the phenotype was examined at the later tail-bud stage (D) Scale bar = 500 μm. The ratio of defective embryos is shown in (E). The numbers on the top indicate the total number of embryos. $n = 3$. (F, G) *X. tropicalis* embryos were injected with *zswim4* CRISPR/Cas9 (200 pg sgRNA plus 0.5 ng Cas9 protein/embryo) or increasing doses of *zswim7* mRNA (50 pg/embryo, 100 pg/embryos). Scale bar = 500 μm. The phenotype was examined at later tail-bud stage. The numbers on the top indicate the total number of embryos. $n = 3$. Data information: *n* indicates biological replicates. Error bars show mean ± standard deviation (SD). Statistical analysis was performed using an unpaired Student's *t* test for (A) and chi-squared test for (C, E, G). *$p < 0.05$, **$p < 0.01$, and ns indicates "not significant" ($p > 0.05$). Source data are available online for this figure.

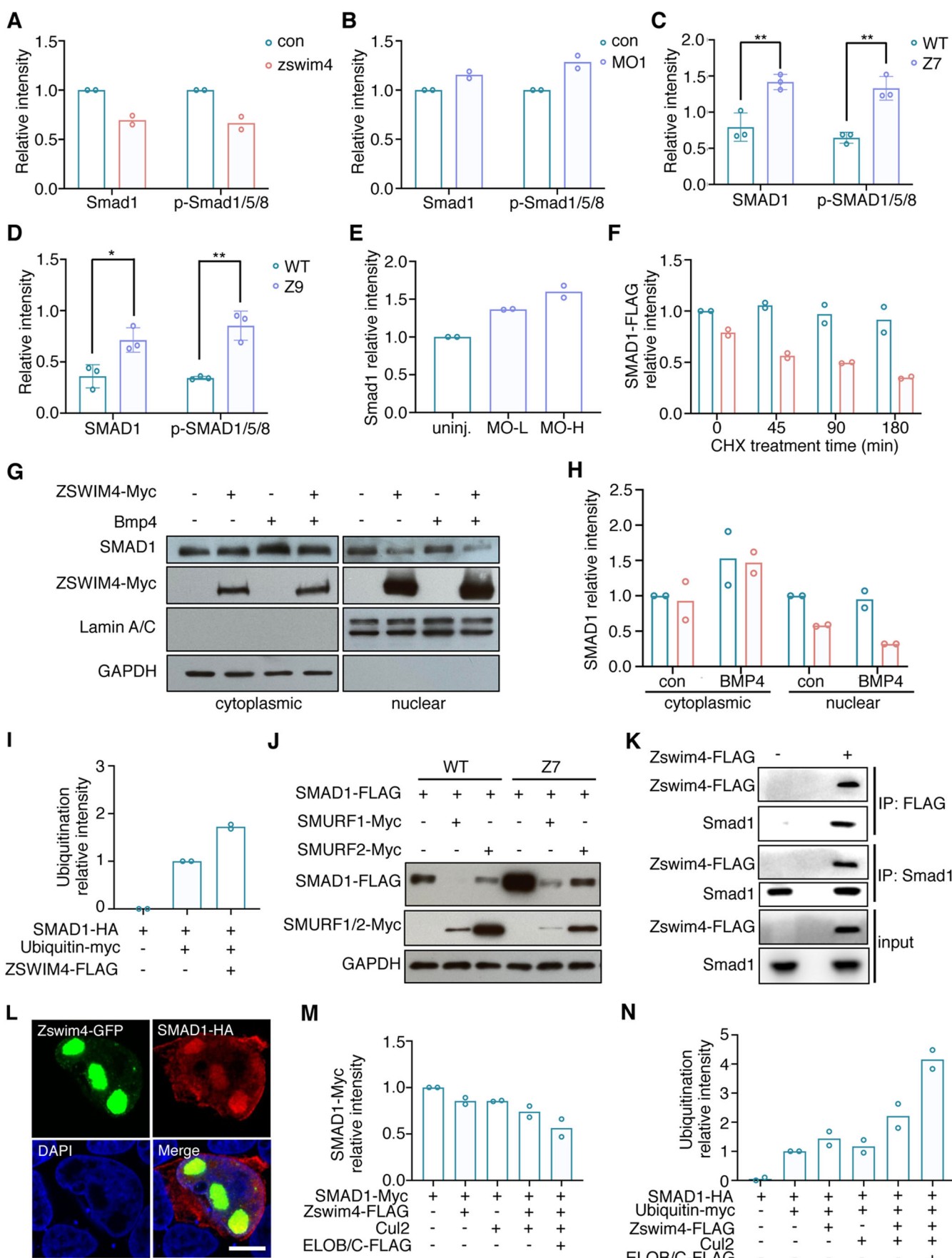

**Figure EV5.  ZSWIM4 reduces SMAD1 protein levels in the nucleus, but not in the cytosol.**

(A, B) Quantifications of Smad1 and p-Smad1/5/8 bands in western blot shown in Fig. 3S (A) and Fig. 3T (B). $n = 2$. (C, D) Quantifications of SMAD1 and p-SMAD1/5/8 bands in western blot shown in Fig. 4K (C) and Fig. 4L (D). $n = 3$. (E) Quantifications of endogenous Smad1 bands in western blot shown in Fig. 5C. $n = 2$. (F) Quantifications of SMAD1-FLAG bands in western blot shown in Fig. 5D. $n = 2$. (G, H) Cell fractionation was performed on HEK293T cells transfected with *ZSWIM4-Myc* (500 ng/ml medium) and treated with BMP4 (100 ng/μl). The cytosolic and nuclear levels of SMAD1 were assayed by western blot. GAPDH and Lamin A/C were used as the loading control for the cytosolic and nuclear proteins, respectively. Quantification of SMAD1 bands is shown in (H). $n = 2$. (I) Quantifications of SMAD1-HA ubiquitination bands in western blot shown in Fig. 5F. $n = 2$. (J) *SMAD1-FLAG* (400 ng/ml medium) was co-transfected with *SMURF1-Myc* (200 ng/ml medium) or *SMURF2-Myc* (200 ng/ml medium) into the *ZSWIM4* mutant cell line, Z7, or wild-type HEK293T cells. The protein level of SMAD1-FLAG was examined by western blot. $n = 2$. One representative blot is shown. (K) Co-IP to detect the interaction between Zswim4-FLAG and endogenous Smad1 in *X. laevis* embryos. $n = 2$. One representative blot is shown. (L) Confocal imaging illustrates the co-localization of Zswim4-GFP and SMAD1-HA in the nucleus of HeLa cells treated with BMP2 (100 ng/μl) for 5 h. Scale bar = 10 μm. $n = 2$. One representative blot is shown. (M) Quantifications of SMAD1-Myc bands in western blot shown in Fig. 7A. $n = 2$. (N) Quantifications of SMAD1-HA ubiquitination bands in western blot shown in Fig. 7B. $n = 2$. Data information: $n$ indicates biological replicates. Error bars show mean ± standard deviation (SD). Statistical analysis was performed using an unpaired Student's $t$ test for (C, D). *$p < 0.05$, **$p < 0.01$. Source data are available online for this figure.

