## [Peer Review File · EMBO Reports]

ZSWIM4 regulates embryonic patterning and BMP signaling by promoting nuclear Smad1 degradation

Chengdong WANG, Ziran Liu, Yelin Zeng, Liangji Zhou, Qi Long, Imtiaz Ul Hassan, Yuanliang Zhang, Xu-Feng Qi, Dong-Qing Cai, Bingyu Mao, Gang Lu, Jianmin Sun, Yong-Gang Yao, Yi Deng, Qian Zhao, Bo Feng, Qin Zhou, Wai-Yee Chan, and Hui Zhao

DOI: [10.15252/embr.202357055](https://doi.org/10.15252/embr.202357055)

Corresponding author(s): Hui Zhao (zhaohui@cuhk.edu.hk)

Review Timeline:

Submission Date:	23rd Feb 23
Editorial Decision:	13th Apr 23
Revision Received:	12th Oct 23
Editorial Decision:	21st Nov 23
Revision Received:	13th Dec 23
Accepted:	15th Dec 23

Editor: Achim Breiling

Transaction Report:

Dear Dr. Zhao,

Thank you for the transfer of your manuscript to EMBO reports. I have now received the reports from the three referees that were asked to evaluate your study, which can be found at the end of this message.

As you will see, the referees indicate that these findings are of interest. However, they have several comments, concerns, and suggestions, indicating that a major revision of the manuscript is necessary to allow publication of the study in EMBO reports. As the reports are below, and all the referee concerns need to be addressed as indicated in the reports, I will not detail them here.

Given the constructive referee comments, I would like to invite you to revise your manuscript with the understanding that all referee concerns must be addressed in the revised manuscript and in a detailed point-by-point response. Acceptance of your manuscript will depend on a positive outcome of a second round of review. It is EMBO reports policy to allow a single round of revision only and acceptance of the manuscript will therefore depend on the completeness of your responses included in the next, final version of the manuscript.

- 1) a .docx formatted version of the final manuscript text (including legends for main figures, EV figures and tables), but without the figures included. Figure legends should be compiled at the end of the manuscript text.
- 2) individual production quality figure files as .eps, .tif, .jpg (one file per figure), of main figures (up to 8) and EV figures. Please upload these as separate, individual files upon re-submission.

- 3) a complete author checklist, which you can download from our author guidelines (<https://www.embopress.org/page/journal/14693178/authorguide>). Please insert page numbers in the checklist to indicate where the requested information can be found in the manuscript. The completed author checklist will also be part of the RPF.

- 4) that primary datasets produced in this study (e.g. RNA-seq, ChIP-seq, structural and array data) are deposited in an appropriate public database. If no primary datasets have been deposited, please also state this in a dedicated section (e.g. 'No primary datasets have been generated and deposited'), see below.

The accession numbers and database should be listed in a formal "Data Availability" section (placed after Materials & Methods) that follows the model below. This is now mandatory (like the COI statement). Please note that the Data Availability Section is restricted to new primary data that are part of this study. This section is mandatory. As indicated above, if no primary datasets have been deposited, please state this in this section

Data availability

8) Regarding data quantification and statistics, please make sure that the number "n" for how many independent experiments were performed, their nature (biological versus technical replicates), the bars and error bars (e.g. SEM, SD) and the test used to calculate p-values is indicated in the respective figure legends (also for potential EV figures and all those in the final Appendix). Please also check that all the p-values are explained in the legend, and that these fit to those shown in the figure. Please provide statistical testing where applicable. Please avoid the phrase 'independent experiment', but clearly state if these were biological or technical replicates. Please also indicate (e.g. with n.s.) if testing was performed, but the differences are not significant. In case n=2, please show the data as separate datapoints without error bars and statistics. See also: <http://www.embopress.org/page/journal/14693178/authorguide#statisticalanalysis>

9) Please add scale bars of similar style and thickness to microscopic images, using clearly visible black or white bars (depending on the background). Please place these in the lower right corner of the images themselves. Please do not write on or near the bars in the image but define the size in the respective figure legend.

10) Please also note our reference format:

12) We now use CRediT to specify the contributions of each author in the journal submission system. CRediT replaces the author contribution section. Please use the free text box to provide more detailed descriptions and remove the author contributions from the manuscript. See also guide to authors: <https://www.embopress.org/page/journal/14693178/authorguide#authorshipguidelines>

Please order the manuscript sections like this, using these names:

Title page - Abstract - Keywords - Introduction - Results - Discussion - Materials and Methods - Data availability section - Acknowledgements - Disclosure and Competing Interests Statement - References - Figure legends - Expanded View Figure legends

Finally, please note that all corresponding authors are required to supply an ORCID ID for their name upon submission of a revised manuscript. Please find instructions on how to link the ORCID ID to the account in our manuscript tracking system in our Author guidelines: <http://www.embopress.org/page/journal/14693178/authorguide#authorshipguidelines>

I look forward to seeing a revised version of your manuscript when it is ready. Please let me know if you have questions or comments regarding the revision.

Yours sincerely,

Referee #1:

The manuscript titled "ZSWIM4 regulates early embryonic patterning and BMP signaling pathway by promoting nuclear Smad1 degradation" by Wang et al. describes a series of experiments that reveal a role for ZSMIM4 in axial patterning of *Xenopus* embryos via regulation of BMP signaling within the nucleus. In this manuscript, the authors show that ZSWIM4 is expressed broadly in the early embryo and is then progressively restricted to the organizer and other dorsal derivatives. The authors show that over expression and knockdown of ZSWIM4 result in dorsalization and ventralization, respectively and that these phenotypes are due to regulation of BMP signaling at the level of SMAD1. Finally, the authors show that ZSWIM4 functions with Cul2 and ELOB/C to ubiquitinate and degrade SMAD1 in the nucleus. This is a novel mechanism and further advances our knowledge on how this important molecular pathway is fine-tuned in the embryo beyond extracellular antagonists like Noggin or Chordin. Overall, this work is interesting and likely to be of great interest to the field and readers of EMBO reports alike. However, there are some points in the manuscript that should be addressed before acceptance for publication.

Major Points:

1. The images of the whole embryo phenotypes following overexpression and knockdown are difficult to assess. Further, the phenotypes do not seem to be consistent. It looks like the embryos failed to complete gastrulation in the moderate and severe phenotypes following *zswim4* RNA injection. Is this true? One would expect a dorsalized phenotype but the blastopore should still close in the event of BMP inhibition. These embryos look less like they're dorsalized than they look like they've not closed their blastopores. The morphants are shortened but seem to have well-patterned heads. Further, the crispant at stage 28 in figure 1I looks very similar to the overexpression phenotype in figure 1A. I would suggest that the authors use a previously standardized method to score dorso-ventral phenotypes such as the Dorsoanterior index (Kao and Elinson 1988 or Monteir et al. 2018). This will be important when comparing the morphants to the crispants (Figure 1F and 1I).
2. These discrepancies make the ZSWIM4 null mutant an important piece of the story yet the authors state that the lack of any phenotype is due to compensation via ZSWIM7. Therefore, I suggest that the authors confirm this. If this model is correct, the *Zswim4* phenotype should be observed by knocking down ZSWIM7 in their ZSWIM4 homozygous mutants.
3. Similarly, the result of obtaining 25% of ventralized embryos when the F0 crispants were crossed with wild-type males (page 7) is confusing. This suggests that a ZSWIM4 mutation is haplo-insufficient. How does this fit with the absence of the phenotype in the homozygous null? Is there no change in the expression level of *zswim7* when one copy of *Zswim4* is mutated. Further, these numbers seem low for an *X. tropicalis* mating, which typically yield far more than 126 (typically over 500 and often over 1000). Further, first time egg lays can be less robust than subsequent ovulations. I suggest that the authors repeat these matings using the same females and compare the percentages. If the same ratios are found, then the embryos with a phenotype should be assayed for *zswim7* as was done in Fig S5E.
4. The authors use ZSWIM4 overexpression to test their hypotheses that it induces dorsal fate at the expense of ventral identity. However, it's not possible to determine if these injections are traced. The authors show restricted expression domains of *sizzled* and *vent1* following *Zswim4* overexpression but does this occur broadly or cell autonomously as would be expected given the proposed mechanisms. Therefore, the authors should repeat the over expression experiments in Figure 2 and include a tracer like LacZ which remains visible following in situ hybridization. This control will add support to their model if they see a cell autonomous loss of ventral gene expression (*sizzled*) and a gain of dorsal fate, as would be expected from the default model (*myoD* or *Sox2*, depending on the germ layer). Also, the secondary axis induction experiments in figure 2U need to be traced and either photographed to show clear secondary axis induction or stained with antibodies that mark dorsal tissue to show ectopic axes (12/101, or Tor70).

Minor points

1. Some of the in situs for Zswim4 should be clearer. Fig 1E is either over stained or over-exposed. Embryos in Fig 1D and 1E should be bisected to show the relative dorsal-ventral differences. Where is the embryo in figure 1I cut for the sections in figure 1J-K?
2. What is the purpose of showing Zswim4 relative expression in figure 4G,H? This should be addressed in the text.
3. The authors should be precise in which *Xenopus* species they show in their figures. The legends should state either *X. laevis* or *X. tropicalis* when appropriate.

Referee #2:

Overall the results finding a new regulator of BMP signaling and dorsoventral patterning is interesting. The manuscript is well written, but there are significant problems with the experiment details provided and the conclusions derived from the results. This paper can be significantly improved with the following suggestions.

1. On page 6, in the second last line authors state, "...the relative number of embryos in the severe and moderate groups is decreased, while the relative number of embryos in the mild and normal groups increased (Figures 2G and 2H)....". The data does not support this conclusion. In Figure 2G, the number of normal embryos decreases, and the number of mild embryos increases, leading to the modest phenotype rescue. Also, in 2H, the increase is in the mild phenotype and not normal embryos. Authors could correct this by changing their wording.
2. Another major concern I have is I do not see the number of times the experiment is repeated for:
 - a. In situ data Fig 2J-O
 - b. qPCR Fig 2P
 - c. western blots. Example Fig 3W and 3X and the number of other experiments.I would like a clear information table for the sample number (n=?) for each experiment and how many times the experiment was repeated for each experiment.
3. For the discrepancy between F0 and F2 embryos, authors can do multiple experiments to support their argument of genetic compensation. First, they could try to rescue the F0 phenotype with zswim7. Also, they could knockdown/knockout zswim7 in F2 embryos to see if zswim7 compensates for zswim4. This will be informative and important to show that knockdown vs. knockout can generate different phenotypes. (not essential)
4. Figure S7 does not support the conclusions made by the authors, similar to my point above about Figures 2G and 2H. On page 10, the authors state that "A decrease in the total SMAD1 level was observed in the nuclear fraction after overexpression of ZSWIM4, compared with the levels in the nuclear fraction of control cells" I do not see much difference. This is, again hard to evaluate because I don't know anything about biological and technical repeats. If the experiment is only done once, then I would not agree with this conclusion. I would like to see the blots for all the experimental repeats.
5. No information about concentrations of sgRNA or MO is given, or at least I could not find it easily. Also, I want biological and technical repeats for each experiment, importantly with CRISPRs, MOs, and rescue.
6. The experiments are designed very well, but they used two models (*Xenopus* and human cell lines), which are related to different nomenclature. In my opinion, gene and protein names are mixed. I know human genes are named in italics and uppercase (e.g., SMAD4), for the protein name should be without italics. In the case of *Xenopus*, it's the same, but only the first letter for the protein is in uppercase (e.g., smad4/Smad4). In a few sentences, I notice it's mixed. For me, it is a bit confusing and incorrect if they only talk about results from the cell lines, but they use nomenclature for *Xenopus*. E.g., "These results suggest that ZSWIM4 mainly reduced the total level of Smad1 and the level of phosphorylated SMAD1/5/8 in the nuclear fraction, which is consistent with its nuclear localization." Results are on human cell lines; I believe Smad1 should be labeled SMAD1. I found this also in other figure legends and text.

Minor comments:

1. Consistency is an issue in many places:
 - a. "...we injected bmp4 mRNA (300 pg/embryo)..." and in the next line "...high dose of zswim4 mRNA (200 pg/embryos)..." If authors want to give us that information in the text, they should do it for every injected mRNA.
 - b. knockdown is sometimes written together or separately.
2. "Sections of the stained embryos also revealed zswim4 signals in the foregut, neural tube, and the region surrounding the notochord" - expression is better than the signal for RNA.
3. "12% of the embryos injected with chordin mRNA (6 of 51) had a secondary axis, while after co-injection of zswim4 mRNA (200 pg/embryo) and chordin mRNA, 29% (23 of 79) of the embryos had a secondary axis (Figures 3U, V)". No information about chordin mRNA concentration was given in the text.
4. Some Western Blots have quantification. It would be nice to see numbers, not only pictures for all western blots, especially when not everywhere endogenous control is equal.

Referee #3:

This paper reports a functional and biochemical analysis of the role of the *zswim4* gene in *Xenopus* embryonic development. The *Zswim* family of proteins has not received much attention so far, in particular in early vertebrate development, and this study represents the first attempt to functionally characterize *zswim4*.

It is proposed that *Zswim4* encodes a nuclear factor involved in BMP pathway regulation, through ubiquitin-mediated degradation of nuclear *Smad1*, in complex with *Cul2/RING* and *Elongin B* and *C*. It was known that *Smad1* stability was controlled by *Smurf1*-mediated degradation. This study thus extends this notion to *Zswim4*, through a distinct molecular pathway.

The paper combines functional experiments in the *Xenopus* embryo, using classical approaches, together with multiple biochemical experiments to reveal the possible mechanism of action of *Zswim4*. While the biochemistry part of the paper appears rather solid, the functional data are of low quality in general. In its present form, the study does not fully establish that the uncovered molecular complex involving *Zswim4* is responsible for its function in the embryo.

Major concerns:

1. Figure 1 reports the spatio-temporal expression of *zswim4* transcripts during *Xenopus* embryogenesis. Wong et al (2016) reported very similar data for a gene then called *zswim5*. The authors must clarify whether *zswim5* in Wong et al. actually corresponds to *zswim4* in the present study. If that is the case, the new data is dispensable or should be displayed in supplement. However, Wong et al did not report maternal expression of *zswim5*, whereas *zswim4* is clearly maternal in Fig. 1B. If *zswim4* and *zswim5* actually represent different members of the same family, it is then necessary to knock them down concomitantly to study their role, as they appear largely co-expressed.
2. The phenotypic classes displayed in Fig. 2A appear arbitrary (mild and moderate are similar) and confusing, as a dorsalizing factor is not expected to repress head formation. It would be preferable to ectopically inject *zswim4* in ventral territories, where it is not expressed and expected to cause secondary axis formation. This assay would also offer a basis to evaluate the cooperation by *Zswim4* molecular partners identified in the study. It is standard procedure to co-inject lineage tracers to better evaluate the consequences on the expression of molecular markers.
3. Again, the morphant phenotypic classes defined in Figure 2F appear arbitrary. It would certainly help to use landmark molecular markers to better characterize those classes. It would make sense to include the prechordal mesoderm marker *gooseoid* among those, in relation to the cyclopia visible in late *crispant* embryos. The manuscript does not report a dose-response evaluation of the *zswim4* morpholinos, such that it is unclear whether the experimental conditions were optimized. The knockdown effects appear rather weak, suggesting that either conditions were not optimal or that compensation by other *Zswim* family members, as suggested by the authors themselves, may take place. It is important to address this issue to fully convince the readers of the importance of this poorly characterized family in embryonic development. The rescue assays are not convincing, as the proportion of normal embryos is not increased.
4. It appears necessary to prove that *Zswim4* protein is actually localized in the nuclei of the dorsal organizer or the neural plate, where its activity is claimed to take place. Conversely, it would strengthen the conclusions to demonstrate that the levels of immunofluorescent p*Smad1* are modified by overexpression and knockdown of *Zswim4*.
5. To firmly establish that the molecular complex identified in biochemical experiments is responsible for *Zswim4* activity in the embryo, the authors must document the presence of its constituents in the embryo, and evaluate their role. In that respect, the data presented in Figure 7E are not fully convincing and lack elementary controls (*Cul2* or *ELOB/C* alone).

Minor concerns:

1. In Fig. 3S it is unclear what the control situation looks like, and why would *Bmp4* repress the expression of its well-characterized positive target *sizzled*?
2. The data shown in Fig. 3U,V is of very poor quality. It is virtually impossible to discern secondary axes; *chordin* effects are very weak. It would actually make more sense to antagonize *Bmp4*-mediated ventralization with *Zswim4*.
3. The data presented in Fig. 4G,H on *Zswim4* expression does not make total sense.
4. The WB in Fig. 4K,L should display quantification of signals.
5. The data presented in Fig. 5C and 3X are not very consistent.
6. The authors argue that the co-localization of *Zswim4*-GFP and *Smad1* in HeLa nuclei (Fig. 5M) supports the view that *Zswim4* causes *Smad1* nuclear degradation. This reviewer does not understand the logic, and at the very least the levels of *Smad1* without *Zswim4* or the evidence that *Zswim4* is not functional in this condition should be presented.

7. The authors suggest that Zswim4 regulates the levels of Smad1 in the nucleus and not its level of phosphorylation in response to Bmp receptor activation. The data of cell fractionation presented in Fig. S7 does not appear totally consistent with this view, and the relative quantification of Smad1 and pSmad1 should be provided.

8. The text contains many typos and needs to be edited to improve English.

Referee #1:

The manuscript titled "ZSWIM4 regulates early embryonic patterning and BMP signaling pathway by promoting nuclear Smad1 degradation" by Wang et al. describes a series of experiments that reveal a role for ZSMIM4 in axial patterning of Xenopus embryos via regulation of BMP signaling within the nucleus. In this manuscript, the authors show that ZSWIM4 is expressed broadly in the early embryo and is then progressively restricted to the organizer and other dorsal derivatives. The authors show that over expression and knockdown of ZSWIM4 result in dorsalization and ventralization, respectively and that these phenotypes are due to regulation of BMP signaling at the level of SMAD1. Finally, the authors show that ZSWIM4 functions with Cul2 and ELOB/C to ubiquitinate and degrade SMAD1 in the nucleus. This is a novel mechanism and further advances our knowledge on how this important molecular pathway is fine-tuned in the embryo beyond extracellular antagonists like Noggin or Chordin. Overall, this work is interesting and likely to be of great interest to the field and readers of EMBO reports alike. However, there are some points in the manuscript that should be addressed before acceptance for publication.

Major Points:

1. The images of the whole embryo phenotypes following overexpression and knockdown are difficult to assess. Further, the phenotypes do not seem to be consistent. It looks like the embryos failed to complete gastrulation in the moderate and severe phenotypes following zswim4 RNA injection. Is this true? One would expect a dorsalized phenotype but the blastopore should still close in the event of BMP inhibition. These embryos look less like they're dorsalized than they look like they've not closed their blastopores. The morphants are shortened but seem to have well-patterned heads. Further, the crispant at stage 28 in figure 1I looks very similar to the overexpression phenotype in figure 1A. I would suggest that the authors use a previously standardized method to score dorso-ventral phenotypes such as the Dorsoanterior index (Kao and Elinson 1988 or Monteir et al. 2018). This will be important when comparing the morphants to the crispants (Figure 1F and 1I).

Re: Thank you very much for your suggestion. Upon injecting 100 pg of zswim4 mRNA per embryo, we observed severe phenotypes, including gastrulation defects and embryonic death at the later neurula stage. Thus, we opted for lower doses of zswim4 mRNA (12.5 pg, 25 pg, 50 pg) to study the overexpression phenotype. The injected embryos showed dorsalized phenotype including shortened and disappeared tails (Fig. 2A, B). We re-injected zswim4MO and the morphants showed an enlarged ventral side, shortened trunk, and suppression of the head that is confirmed by the reduced *otx2* expression (Fig. 2C-E). The morphant phenotype is similar to that of crispants (Fig. 2F). Following the reviewer's suggestion, we have incorporated the Dorsoanterior index to score the phenotypes (Fig. 2A, C, and F).

2. These discrepancies make the ZSWIM4 null mutant an important piece of the story yet the authors state that the lack of any phenotype is due to compensation via ZSWIM7. Therefore, I suggest that the authors confirm this. If this model is correct, the Zswim4 phenotype should be observed by knocking down ZSWIM7 in their ZSWIM4 homozygous mutants.

Re: Thank you very much for your suggestions. We proceeded to inject zswim7MO into zswim4F2 embryos (offsprings of F1 generation). After knockdown of zswim7 in zswim4F2 embryos, about half of them showed defects in tail and head regions (Fig. EV4B, C). Notably, when the wild-type embryos were injected with the same dose of zswim7MO, they mostly showed normal morphology. Furthermore, we conducted rescue experiments in zswim4 crispants by co-injecting zswim7 mRNA. The phenotypes of zswim4 crispants were partially rescued by zswim7 in a dose-dependent manner (Fig. EV4D, E). These results strongly suggest that up-regulated Zswim7 contributes to the genetic compensation in zswim4 mutants.

3. Similarly, the result of obtaining 25% of ventralized embryos when the F0 crispants were crossed with wild-type males (page 7) is confusing. This suggests that a ZSWIM4 mutation is haplo-insufficient. How does this fit with the absence of the phenotype in the homozygous null? Is there no change in the expression level of zswim7 when one copy of Zswim4 is mutated. Further, these numbers seem low for an X. tropicalis mating, which typically yield far more than 126 (typically over 500 and often over 1000). Further, first time egg lays can be less robust than subsequent ovulations. I suggest that the authors repeat these matings using the same females and compare the percentages. If the same ratios are found, then the embryos with a phenotype should be assayed for zswim7 as was done in Fig S5E.

Re: We appreciate the constructive suggestion. However, we don't have the F0 crispant frog right now. The F0 crispant frogs were established 6 years ago. Due to the space limit, the F0 crispant frogs were discontinued after we obtained F1 zswim4 KO frog line. It would be much appreciated if you consider exempting us from doing this experiment. zswim4 has a strong maternal expression in the animal hemisphere, and we thought that the ventralization phenotype observed in F1 embryos (Fig. 2H, I) may be largely due to the lack of maternal zswim4.

4. The authors use ZSWIM4 overexpression to test their hypotheses that it induces dorsal fate at the expense of ventral identity. However, it's not possible to determine if these injections are traced. The authors show restricted expression domains of sizzled and vent1 following Zswim4 overexpression but does this occur broadly or cell autonomously as would be expected given the proposed mechanisms. Therefore, the authors should repeat the over expression experiments in Figure 2 and include a tracer like LacZ which remains visible following in situ hybridization. This control will add support to their model if they see a cell autonomous loss of ventral gene expression (sizzled) and a gain of dorsal fate, as would be expected from the default model (myoD or Sox2, depending on the germ layer). Also, the secondary axis induction experiments in figure 2U need to be traced and either photographed to show clear secondary axis induction or stained with antibodies that mark dorsal tissue to show ectopic axes (12/101, or Tor70).

Re: Thank you very much for your suggestion. We have repeated the in situ hybridization using embryos injected with lacZ mRNA and either zswim4 mRNA or MO (Fig 3A-G). LacZ staining was employed to trace the injected zswim4 mRNA or MO. In the secondary axis induction experiment, the chordin and zswim4 mRNAs were also co-injected with lacZ mRNA.

After lacZ staining, the embryos were stained with 12/101 antibody to visualize the body axis (Fig 3Q, R).

Minor points

1. *Some of the in situs for Zswim4 should be clearer. Fig 1E is either over stained or over-exposed. Embryos in Fig 1D and 1E should be bisected to show the relative dorsal-ventral differences. Where is the embryo in figure 1I cut for the sections in figure 1J-K?*

Re: We replaced the previous Fig 1E with an image of normal stained embryos (Fig. EV1F). Bisected embryos (previously in Fig 1D and 1E) are shown in Fig 1C. The embryonic region cut for sections in Fig 1J-K is illustrated in Fig. EV1I.

2. *What is the purpose of showing Zswim4 relative expression in figure 4G,H? This should be addressed in the text.*

Re: Thank you for asking this question. We aimed to determine whether ZSWIM4 mRNA level is altered in the mutant cell lines. Since ZSWIM4 mRNA was not reduced, we opted to exclude this data from the revised figure.

3. *The authors should be precise in which Xenopus species they show in their figures. The legends should state either X. laevis or X. tropicalis when appropriate.*

Re: Thank you very much for your suggestions. We have stated specifically *X. laevis* or *X. tropicalis* in legends.

----- Referee #2:

Overall the results finding a new regulator of BMP signaling and dorsoventral patterning is interesting. The manuscript is well written, but there are significant problems with the experiment details provided and the conclusions derived from the results. This paper can be significantly improved with the following suggestions.

1. *On page 6, in the second last line authors state, "...the relative number of embryos in the severe and moderate groups is decreased, while the relative number of embryos in the mild and normal groups increased (Figures 2G and 2H)...". The data does not support this conclusion. In Figure 2G, the number of normal embryos decreases, and the number of mild embryos increases, leading to the modest phenotype rescue. Also, in 2H, the increase is in the mild phenotype and not normal embryos. Authors could correct this by changing their wording.*

Re: Thank you very much for highlighting this point. We have repeated the MO rescuing experiments, which showed that co-injection of *zswim4* can evidently reduce the ratio of severe phenotype, while concurrently increasing the proportion of those presenting with a normal phenotype (Fig. 2E). Our conclusion remains the same.

2. Another major concern I have is I do not see the number of times the experiment is repeated for: a. In situ data Fig 2J-O
b. qPCR Fig 2P
c. western blots. Example Fig 3W and 3X and the number of other experiments.

I would like a clear information table for the sample number (n=?) for each experiment and how many times the experiment was repeated for each experiment.

Re: Thank you very much for bringing up this question. All the experiments were repeated at least twice. These include the in situ hybridization in Fig 2J-O, the qPCR in Fig 2P and the western blot in Fig 3W and 3X. We have shown the qPCR and luciferase results using the column with a scatter plot to visualize the data points. The numbers of embryos for in situ hybridization and phenotype study were mentioned in the figure legends. Additionally, replicate data for the qPCR, luciferase assay, and western blot can be accessed in the source data we've provided with our submission.

3. *For the discrepancy between F0 and F2 embryos, authors can do multiple experiments to support their argument of genetic compensation. First, they could try to rescue the F0 phenotype with zswim7. Also, they could knockdown/knockout zswim7 in F2 embryos to see if zswim7 compensates for zswim4. This will be informative and important to show that knockdown vs. knockout can generate different phenotypes. (not essential)*

Re: Thank you very much for your suggestion. We proceeded with injection of zswim7MO to zswim4F2 embryos. After knockdown of zswim7 in zswim4F2 embryos, about half of them showed defects in tail and head regions (Fig. EV4B, C), while the wild type embryos injected with same dose of zswim7MO predominantly maintained their normal phenotype. We also did rescue experiments in zswim4 crispants by co-injecting zswim7 mRNA. The phenotype of zswim4 crispants was partially rescued by injection of zswim7, with the degree of rescue being dose-dependent (Fig. EV4D, E). These results strongly suggest that up-regulated zswim7 plays an important role in the genetic compensation observed in zswim4 mutants.

4. *Figure S7 does not support the conclusions made by the authors, similar to my point above about Figures 2G and 2H. On page 10, the authors state that "A decrease in the total SMAD1 level was observed in the nuclear fraction after overexpression of ZSWIM4, compared with the levels in the nuclear fraction of control cells" I do not see much difference. This is, again hard to evaluate because I don't know anything about biological and technical repeats. If the experiment is only done once, then I would not agree with this conclusion. I would like to see the blots for all the experimental repeats.*

Re: Thank you very much for highlighting this point. In response, we repeated the cell fractionation experiment. The results showed the SMAD1 protein was evidently reduced in the nucleus upon transfection of ZSWIM4. However, no such reduction was observed in the cytosol (as shown in Fig. EV5A). Please note that this experiment was conducted with two independent biological replicates.

5. No information about concentrations of sgRNA or MO is given, or at least I could not find it easily. Also, I want biological and technical repeats for each experiment, importantly with CRISPRs, MOs, and rescue.

Re: Thank you very much for highlighting this point. We added the concentrations of sgRNA and MO, as well as the numbers of biological repeats of CRISPRs, MOs, and rescue in our revised manuscript. The information can be found in the figure legends.

6. The experiments are designed very well, but they used two models (*Xenopus* and human cell lines), which are related to different nomenclature. In my opinion, gene and protein names are mixed. I know human genes are named in italics and uppercase (e.g., *SMAD4*), for the protein name should be without italics. In the case of *Xenopus*, it's the same, but only the first letter for the protein is in uppercase (e.g., *smad4/Smad4*). In a few sentences, I notice it's mixed. For me, it is a bit confusing and incorrect if they only talk about results from the cell lines, but they use nomenclature for *Xenopus*. E.g., "These results suggest that ZSWIM4 mainly reduced the total level of *Smad1* and the level of phosphorylated *SMAD1/5/8* in the nuclear fraction, which is consistent with its nuclear localization." Results are on human cell lines; I believe *Smad1* should be labeled *SMAD1*. I found this also in other figure legends and text.

Re: Thank you very much for bringing up this point. We have carefully checked the text and figure legends and corrected them.

Minor comments:

1. Consistency is an issue in many places:

a. "...we injected *bmp4* mRNA (300 pg/embryo)..." and in the next line "...high dose of *zswim4* mRNA (200 pg/embryos)..." If authors want to give us that information in the text, they should do it for every injected mRNA.

b. knockdown is sometimes written together or separately.

Re: Thank you very much for your suggestions. We have carefully checked the manuscript and listed the concentration and injection doses following this rule.

2. "Sections of the stained embryos also revealed *zswim4* signals in the foregut, neural tube, and the region surrounding the notochord" - expression is better than the signal for RNA.

Re: Thank you very much. We have changed it accordingly.

3. "12% of the embryos injected with *chordin* mRNA (6 of 51) had a secondary axis, while after co-injection of *zswim4* mRNA (200 pg/embryo) and *chordin* mRNA, 29% (23 of 79) of the embryos had a secondary axis (Figures 3U, V)". No information about *chordin* mRNA concentration was given in the text.

Re: Thank you very much for your suggestion. We added the *chordin* mRNA concentration.

4. Some Western Blots have quantification. It would be nice to see numbers, not only pictures for all western blots, especially when not everywhere endogenous control is equal.

Re: Thank you very much. We have done quantification for all the western blots.

----- Referee #3:

This paper reports a functional and biochemical analysis of the role of the zswim4 gene in Xenopus embryonic development. The Zswim family of proteins has not received much attention so far, in particular in early vertebrate development, and this study represents the first attempt to functionally characterize zswim4.

It is proposed that Zswim4 encodes a nuclear factor involved in BMP pathway regulation, through ubiquitin-mediated degradation of nuclear Smad1, in complex with Cul2/RING and Elongin B and C. It was known that Smad1 stability was controlled by Smurf1-mediated degradation. This study thus extends this notion to Zswim4, through a distinct molecular pathway.

The paper combines functional experiments in the Xenopus embryo, using classical approaches, together with multiple biochemical experiments to reveal the possible mechanism of action of Zswim4. While the biochemistry part of the paper appears rather solid, the functional data are of low quality in general. In its present form, the study does not fully establish that the uncovered molecular complex involving Zswim4 is responsible for its function in the embryo.

Major concerns:

1. Figure 1 reports the spatio-temporal expression of zswim4 transcripts during Xenopus embryogenesis. Wong et al (2016) reported very similar data for a gene then called zswim5. The authors must clarify whether zswim5 in Wong et al. actually corresponds to zswim4 in the present study. If that is the case, the new data is dispensable or should be displayed in supplement. However, Wong et al did not report maternal expression of zswim5, whereas zswim4 is clearly maternal in Fig. 1B. If zswim4 and zswim5 actually represent different members of the same family, it is then necessary to knock them down concomitantly to study their role, as they appear largely co-expressed.

Re: Thank you very much for raising this question. We have recently accomplished a study that classifies the Zswim protein family genes in *Xenopus* (Zool Res. 2023 44(3): 663-674. doi: 10.24272/j.issn.2095-8137.2022.418). The zswim5 in Wong et al (2016) is the zswim4 in this study. We have displayed the detailed in situ hybridization data for zswim4 in the supplementary figure.

2. The phenotypic classes displayed in Fig. 2A appear arbitrary (mild and moderate are similar) and confusing, as a dorsalizing factor is not expected to repress head formation. It would be preferable to ectopically inject zswim4 in ventral territories, where it is not expressed and expected to cause secondary axis formation. This assay would also offer a basis to

evaluate the cooperation by Zswim4 molecular partners identified in the study. It is standard procedure to co-inject lineage tracers to better evaluate the consequences on the expression of molecular markers.

Re: Thank you very much for your insightful suggestion. The injection of 100 pg *zswim4* mRNA per embryo caused severe phenotypes, including gastrulation defects and embryonic death at later neurula stage. It suggests that *zswim4* has a significant impact on embryonic development and its functions may extend beyond BMP signaling. Thus, we reduced the injection doses of *zswim4* mRNA (12.5 pg/embryo, 25 pg/embryo, and 50 pg/embryo) to study the phenotype. The injected embryos showed a dorsalized phenotype, notably with shortened and even absent tail (Fig 2A, B). When we injected *zswim4* and its CRL partners into the ventral regions, no induced secondary axis was observed. This leads us to believe that Zswim4 serves as a fine-tune modulator of BMP signaling, and it does not have the capability to induce secondary body axis. Moreover, increasing the dose of *zswim4* could result in adverse effects, leading to embryonic death. Following your suggestion, we have co-injected *zswim4* mRNA with *lacZ* mRNA into ventral blastomeres and assessed the expression of ventral mesoderm marker genes, i.e., *vent1*, *vent2*, and *sizzled* (Fig 3A-C).

3. Again, the morphant phenotypic classes defined in Figure 2F appear arbitrary. It would certainly help to use landmark molecular markers to better characterize those classes. It would make sense to include the prechordal mesoderm marker goosecoid among those, in relation to the cyclopia visible in late crispant embryos. The manuscript does not report a dose-response evaluation of the zswim4 morpholinos, such that it is unclear whether the experimental conditions were optimized. The knockdown effects appear rather weak, suggesting that either conditions were not optimal or that compensation by other Zswim family members, as suggested by the authors themselves, may take place. It is important to address this issue to fully convince the readers of the importance of this poorly characterized family in embryonic development. The rescue assays are not convincing, as the proportion of normal embryos is not increased.

Re: Thank you very much for bringing up this matter. In response, we re-injected *zswim4*MO1 using different doses to further study the phenotypes (Fig. 2E). We used the expression of brain marker *otx2* to differentiate and characterize the phenotype categories (Fig. 2D). Additionally, the rescue experiment was performed (Fig. 2E). Notably, *goosecoid* expression is reduced after the knockdown of *zswim4* (Fig. 3F, I).

4. It appears necessary to prove that Zswim4 protein is actually localized in the nuclei of the dorsal organizer or the neural plate, where its activity is claimed to take place. Conversely, it would strengthen the conclusions to demonstrate that the levels of immunofluorescent pSmad1 are modified by overexpression and knockdown of Zswim4.

Re: Thank you very much for your suggestions. We performed immunofluorescence staining to examine endogenous Zswim4 protein in cryosectioned embryos. Zswim4 is predominantly localized in the nucleus, and higher Zswim4 signaling density was on the dorsal region at mid-gastrula stage (Fig 1E). We also tried immunofluorescence staining using two anti-p-Smad1/5/8

antibodies (Abcam, #ab92698; CST, #13820S), but could not detect the convincing fluorescence signals in *Xenopus* embryos.

5. To firmly establish that the molecular complex identified in biochemical experiments is responsible for Zswim4 activity in the embryo, the authors must document the presence of its constituents in the embryo, and evaluate their role. In that respect, the data presented in Figure 7E are not fully convincing and lack elementary controls (Cul2 or ELOB/C alone).

Re: Thank you very much for your suggestion. We have examined the spatial expression patterns of *cul2*, *elob* and *eloc* in *Xenopus* embryos. Although their signals were ubiquitous except the yolk plug at stage 10, they were expressed in the dorsal ectoderm and dorsal blastopore at gastrula stages (Fig. S7). Moreover, we repeated the animal cap assay in Fig 7E by incorporating additional groups for Cul2 and ELOB/C (Fig. 7D). When *zswim4*, *cul2* and *elob/c* were co-injected, we found a synergistic enhancement in neural induction (Fig. 7E). More, our luciferase assay showed that, in whole embryos, co-injection of *zswim4*, *cul2*, *elob/c* can synergistically reduce the BRE-luciferase activity (Fig. 7F). These findings strongly suggest that Zswim4 cooperates with Cul2 and ELOB/C to repress BMP signaling in *Xenopus* embryos.

Minor concerns:

1. In Fig. 3S it is unclear what the control situation looks like, and why would Bmp4 repress the expression of its well- characterized positive target sizzled?

Re: Thank you very much. We add a new control embryo (Fig. 3O).

2. The data shown in Fig. 3U,V is of very poor quality. It is virtually impossible to discern secondary axes; chordin effects are very weak. It would actually make more sense to antagonize Bmp4-mediated ventralization with Zswim4.

Re: Thank you very much for bringing up this issue. We have performed immunofluorescence staining using 12/101 antibody to effectively visualize the secondary axis (Fig. 3Q). Moreover, we also used Zswim4 to counteract BMP4-mediated ventralization, and our observations indicated that *zswim4* co-expression alleviated the ventralization effect (Fig. 3M, N).

3. The data presented in Fig. 4G,H on Zswim4 expression does not make total sense.

Re: Thank you very much for your suggestion. We removed Zswim4 expression from the Fig. 4G, H.

4. The WB in Fig. 4K,L should display quantification of signals.

Re: Thank you very much for your suggestion. We have added quantification of signals in Fig 4K, L.

5. The data presented in Fig.5C and 3X are not very consistent.

Re: Thank you very much for bringing up this matter. We have repeated the western blot in Fig 3X, and now the western blot is shown in Fig. 3T.

6. The authors argue that the co-localization of Zswim4-GFP and Smad1 in HeLa nuclei (Fig. 5M) supports the view that Zswim4 causes Smad1 nuclear degradation. This reviewer does not understand the logic, and at the very least the levels of Smad1 without Zswim4 or the evidence that Zswim4 is not functional in this condition should be presented.

Re: Thank you very much for bringing up this matter. We have removed the sentence that “co-localization of Zswim4-GFP and Smad1 in HeLa nuclei supports that Zswim4 causes Smad1 nuclear degradation”. While our co-IP results highlighted a physical interaction between Zswim4 and Smad1, we sought to further investigate their co-localization in HeLa cells. Given that zswim4 cooperates with CRL components to degrade Smad1 protein, and overexpression of zswim4 alone does not entirely deplete the Smad1 protein, it is possible to observe the colocalization of the functional Zswim4 and Smad1. Indeed, the result showed that Zswim4 and Smad1 were colocalized in specific regions of the nucleus (Fig. EV5D).

7. The authors suggest that Zswim4 regulates the levels of Smad1 in the nucleus and not its level of phosphorylation in response to Bmp receptor activation. The data of cell fractionation presented in Fig. S7 does not appear totally consistent with this view, and the relative quantification of Smad1 and pSmad1 should be provided.

Re: Thank you very much for your suggestion. We repeated the cell fractionation assay (Fig. EV5A).

8. The text contains many typos and needs to be edited to improve English.

Re: We carefully checked the manuscript and tried our best to correct typos.

Dear Dr. Zhao,

Thank you for the submission of your revised manuscript to our editorial offices. I have now received the reports from the three referees that have been asked to re-evaluate your paper, you will find below. As you will see, now support the publication of the study in EMBO reports. However, referees #2 and #3 have remaining concerns and suggestions to improve the manuscript, I ask you to address in a final revised manuscript. Please also provide a final p-b-p-response regarding these points.

Moreover, I have these editorial requests I ask you to address:

- Please provide a final title with not more than 100 characters including spaces.
- Please reduce the number of keywords to 5.
- Please provide the abstract written in present tense throughout.
- Please remove the referee access from the 'Data Availability' section (DAS), provide a direct link to the dataset and make sure this is public latest upon publication of the manuscript.
- Please add scale bars of similar style and thickness to all the microscopic images (main, EV and Appendix figures), using clearly visible black or white bars (depending on the background). Please place these in the lower right corner of the images themselves. Please do not write on or near the bars in the image but define the size in the respective figure legend. Presently many images lack scale bars, in particular all those showing embryos.
- Please make sure that the number "n" for how many independent experiments were performed, their nature (biological versus technical replicates), the bars and error bars (e.g. SEM, SD) and the test used to calculate p-values is indicated in the respective figure legends (for main, EV and Appendix figures) of the final revised manuscript. Please also check that all the p-values are explained in the legend, and that these fit to those shown in the figure. Please provide statistical testing where applicable. Please avoid the phrase 'independent experiment', but clearly state if these were biological or technical replicates. Please also indicate (e.g. with n.s.) if testing was performed, but the differences are not significant. In case n=2, please show the data as separate datapoints without error bars and statistics. See also:
<http://www.embopress.org/page/journal/14693178/authorguide#statisticalanalysis>

If n<5, please show single datapoints for diagrams. There are still panels with diagrams without statistics. Please check.

- Please format the figure legends according to our journal style. See the respective section in our guide to authors (please find the link below). Please separate each panel description by a line brake and make sure that the panels are listed in alphabetic order. Moreover, please add to each legend a 'Data Information' section explaining the statistics used or providing information regarding replicates and scales.

In addition, I would need from you:

Best,

Referee #1:

My concerns have been addressed

Referee #2:

The authors have made significant changes to the manuscript following major and minor revisions, which the reviewer certainly appreciates. However, a few minor concerns remain. Concentrations of mRNA, on many occasions, have not been provided. In the figure EV5, nuclear/cytoplasmic extraction, how much ZSWIM4-Myc was injected? This conc is not present in the legends or text. And the clear decrease in nuclear fraction seen in the figure EV5, why that can not be seen in the previous version of the paper? Is that because the authors increased the concentration of ZSWIM4? How much was injected before vs. in this experiment? Also, how does that conc. relates with previous concentrations used? These are exactly the reasons why accurate information about CRISPR, MO, mRNA, DNA injection, and transfection concentrations need to be provided and kept constant to compare throughout experiments.

The second concern is not all western blots have quantification listed. If the experiment is repeated 2-3 times, then there should be a bar graph with individual data points associated with the western blot to show what individual data looks like and how much the variation is.

Referee #3:

The authors have made major efforts to address my concerns, and those of the other reviewers, several of which were convergent.

This revised version is much improved, and the conclusions further substantiated.

However, I still see a few issues that needs to be addressed.

1. The authors confirmed that zswim4 is the same gene that was studied in Wong et al, and called zswim5 back then. However, Wong et al did not report maternal expression, whereas Wang et al in the present study do detect massive maternal mRNA and protein expression. Please clarify.
2. The new experiments indeed suggest functional compensation by zswim members in zswim4 mutants. However, to substantiate this claim, the authors need to demonstrate that zswim7, which they selected to evaluate compensation, is indeed up-regulated in the SMO and the neural plate, where it could replace zswim4.
3. In response to my comment, the authors introduced an anti-zswim4 antibody to confirm the enrichment of the protein in dorsal nuclei. However, they do not describe the origin of this antibody. If it is the first report using this reagent it must be validated in this study.
4. Ventral injection of zswim4 does not appear sufficient to induce an ectopic body axis. In their response to my comment, the authors explain that zswim4 would fine-tune Smad1 signalling levels, presumably in complement to Smurf and other regulators. I concur to this balanced view and suggest that it is made more apparent in the discussion, and perhaps in the title.
5. Typos :
p4: mechanically should be mechanistically
p7: ...DAI 6 (Fig 2F). The authors meant DAI 4. However, it is more DAI 3 to me.
p7: ...heterogeneous (Fig 2H and 2I). The authors probably meant heterozygous ?
p8: Knockdown of zswim7 in zswim4F2 evidently improved the ratio of defective embryos... the verb improve is not suitable here.

Dear Dr. Achim Breiling,

We would like to express our gratitude to both you and the reviewers for providing valuable insights and constructive suggestions regarding our early submission entitled "ZSWIM4 regulates embryonic patterning and BMP signaling by promoting nuclear Smad1 degradation" (EMBOR-2023-57055V2). We have carefully reviewed the comments from Reviewer 2 and 3 and are committed to conducting additional experiments to address the questions they raised.

Our point-by-point response to the reviewers' feedback is provided in the attached pages. We have addressed all the reviewers' comments by performing new experiments and revising the manuscript accordingly. Furthermore, we have implemented the suggested editorial amendments based on your guidelines.

Best regards,

Hui Zhao Ph.D.

- Please provide a final title with not more than 100 characters including spaces.

Answer: We have provided a new title "ZSWIM4 regulates embryonic patterning and BMP signaling by promoting nuclear Smad1 degradation".

- Please reduce the number of keywords to 5.

Answer: We have reduced the number of the keywords to 5.

- Please provide the abstract written in present tense throughout.

Answer: We changed and used the present tense for our Abstract.

- Please remove the referee access from the 'Data Availability' section (DAS), provide a direct link to the dataset and make sure this is public latest upon publication of the manuscript.

Answer: The data can be accessed by the public once we get the PubMedID or DOI.

-Please add scale bars of similar style and thickness to all the microscopic images

Answer: We have added scale bars with the similar style and thickness to all the microscopic images.

- Please make sure that the number "n" for how many independent experiments were performed, their nature

Answer: We have done it. The number "n" indicated the repeats of independent experiments.

In addition, I would need from you:

- a short, two-sentence summary of the manuscript (not more than 35 words).

Answer: Please find the summary below.

Zswim4 fine-tunes BMP signaling in *Xenopus* early embryonic patterning by interacting with Cullin2, Elongin B, and C, forming a protein disruption complex that ultimately reduces nuclear Smad1 protein stability.

- two to four short (!) bullet points highlighting the key findings of your study (two lines each).

Answer: Please find the bullet points.

- Zswim4 is a novel inhibitor of the BMP signaling pathway, localized in the nucleus.
- Zswim4 physically interacts with Smad1 through the Smad1 MH1 domain.
- Zswim4 reduces Smad1 stability and increases Smad1 ubiquitination in the nucleus.
- Zswim4 interacts with Cullin2 and Elongin B and C, forming a protein disruption complex to promote Smad1 ubiquitination.

- a schematic summary figure as separate file that provides a sketch of the major findings (not a data image) in jpeg or tiff format (with the exact width of 550 pixels and a height of not more than 400 pixels) that can be used as a visual synopsis on our website.

Answer: Please find the schematic summary in the attached file (Schematic summary figure).

Referee #2:

(1) The authors have made significant changes to the manuscript following major and minor revisions, which the reviewer certainly appreciates. However, a few minor concerns remain. Concentrations of mRNA, on many occasions, have not been provided. In the figure EV5, nuclear/cytoplasmic extraction, how much ZSWIM4-Myc was injected? This conc is not present in the legends or text. And the clear decrease in nuclear fraction seen in the figure EV5, why that can not be seen in the previous version of the paper? Is that because the authors increased the concentration of ZSWIM4? How much was injected before vs. in this experiment? Also, how does that conc. relates with previous concentrations used? These are exactly the reasons why accurate information about CRISPR, MO, mRNA, DNA injection, and transfection concentrations need to be provided and kept constant to compare throughout experiments.

Answers: Thank you very much for raising this concern. We have checked the manuscript and added the concentrations of mRNA.

In the nuclear/cytoplasmic extraction experiment (Fig EV5G) in the updated manuscript we increased the concentration of Zswim4-Myc to 500 ng/ml medium compared to the previous experiment, where it was 300 ng/ml medium. Additionally,

we used a new batch of HEK293T cells for this experiment. These adjustments in plasmid transfection amount and cell batch enabled us to observe a noticeable reduction in the nuclear fraction of Zswim4-Myc.

(2) The second concern is not all western blots have quantification listed. If the experiment is repeated 2-3 times, then there should be a bar graph with individual data points associated with the western blot to show what individual data looks like and how much the variation is.

Answers: Thank you very much for your suggestions. We incorporated your suggestions and generated bar graphs for each western blot analysis. These bar graphs are now shown in Figure EV5 of the manuscript. According to EMBO Reports policy, in case $n = 2$, the data should be shown as separate datapoints without error bars and statistics. Therefore, we kindly request permission to present our data, including Figure EV2G, EV5A, EV5B, EV5E, EV5F, EV5H, EV5I, EV5M and EV5N, following the guidelines of EMBO Reports. Additionally, we have incorporated the standard deviation (SD) values and conducted statistical analysis, which are included in the source data of the figures mentioned above, and we have submitted it for your review.

Referee #3:

(1) The authors confirmed that zswim4 is the same gene that was studied in Wong et al, and called zswim5 back then. However, Wong et al did not report maternal expression, whereas Wang et al in the present study do detect massive maternal mRNA and protein expression. Please clarify.

Answers: Thank you for addressing the concern. It appears that Mr. Wong initially did not observe the maternal expression of zswim4, likely due to a low concentration of the antisense RNA probe or insufficient color development time. However, the strong maternal expression of zswim4 was later detected using RT-PCR. To validate this finding further, we extended the NBT/BCIP staining time during whole-mount in situ hybridization, which confirmed the presence of maternal zswim4 expression.

(2) The new experiments indeed suggest functional compensation by zswim members in zswim4 mutants. However, to substantiate this claim, the Cauthors need to demonstrate that zswim7, which they selected to evaluate compensation, is indeed up-regulated in the SMO and the neural plate, where it could replace zswim4.

Answers: Thank you very much for your suggestions. We examined the expression of zswim7 in either WT embryos, zswim4 mutant embryos (zswim4F2) or zswim4 crispants. The results were shown in Fig EV4B and EV4C. Up-regulated expression of zswim7 was observed in about half of zswim4 mutant embryos (57% of zswim4F2). This up-regulation was found to be enriched in the dorsal region, including the SMO and the forming neural plate. Importantly, no up-regulation of zswim7 expression was observed in WT embryos or zswim4 crispants.

(3) In response to my comment, the authors introduced an anti-zswim4 antibody to confirm the enrichment of the protein in dorsal nuclei. However, they do not describe

the origin of this antibody. If it is the first report using this reagent it must be validated in this study.

Answers: Thank you very much for your suggestion. We have added the Zswim4 antibody information in Appendix Table S3, and mentioned that in the Materials and Methods section (Proteins, antibodies, and siRNAs in page 21). We have done a western blot to validate the specificity of anti-Zswim4 antibodies. The western blot was shown in Figure EV2F and EV2G, in which the endogenous Zswim4 protein levels were reduced in *Xenopus* embryo injected with zswim4MO1 or zswim4MO2. This result confirms the specificity of the Zswim4 antibody.

Additionally, we repeated the immunostaining experiments and included rabbit IgG as a negative control, as depicted in Figure EV1M. These experiments served to verify the specificity of the Zswim4 antibody in *Xenopus* embryos and demonstrated that the immunostaining observed was not due to non-specific binding.

(4) Ventral injection of zswim4 does not appear sufficient to induce an ectopic body axis. In their response to my comment, the authors explain that zswim4 would fine-tune Smad1 signaling levels, presumably in complement to Smurf and other regulators. I concur with this balanced view and suggest that it is made more apparent in the discussion and perhaps in the title.

Answers: Thank you very much for your suggestion. We have added the following contents in our Discussion (The end of the first paragraph in page 17).

“Injection with zswim4 into the ventral region of the *Xenopus* embryo did not induce the secondary axis. This observation suggests that Zswim4 serves as a fine-tuned modulator of BMP signaling. It is likely that Zswim4 acts in coordination with other strong regulators to achieve precise and accurate regulation of BMP signaling during embryonic patterning.”

(5) Typos :

p4: mechanically should be mechanistically

p7: ...DAI 6 (Fig 2F). The authors meant DAI 4. However, it is more DAI 3 to me.

p7: ...heterogeneous (Fig 2H and 2I). The authors probably meant heterozygous?

p8: Knockdown of zswim7 in zswim4F2 evidently improved the ratio of defective embryos... the verb improve is not suitable here.

Answers: Thank you very much for pointing out the typos. We have corrected the typos.

Dr. Hui Zhao
The Chinese University of Hong Kong
School of Biomedical Sciences
The Chinese University of Hong Kong
Hong Kong 0000
Hong Kong Special Administrative Region of the People's Republic of China

Dear Dr. Zhao,

I am very pleased to accept your manuscript for publication in the next available issue of EMBO reports. Thank you for your contribution to our journal.

Yours sincerely,
